# A novel enhancer regulates MGMT expression and promotes temozolomide resistance in glioblastoma

Xiaoyue Chen[1], Minjie Zhang[2], Haiyun Gan[2], Heping Wang[3], Jeong-Heon Lee[1], Dong Fang[2], Gaspar J. Kitange[4], Lihong He[4], Zeng Hu[4], Ian F. Parney[5], Fredric B. Meyer[5], Caterina Giannini[6], Jann N. Sarkaria[4] & Zhiguo Zhang[2]

Temozolomide (TMZ) was used for the treatment of glioblastoma (GBM) for over a decade, but its treatment benefits are limited by acquired resistance, a process that remains incompletely understood. Here we report that an enhancer, located between the promoters of marker of proliferation Ki67 (*MKI67*) and O6-methylguanine-DNA-methyltransferase (*MGMT*) genes, is activated in TMZ-resistant patient-derived xenograft (PDX) lines and recurrent tumor samples. Activation of the enhancer correlates with increased MGMT expression, a major known mechanism for TMZ resistance. We show that forced activation of the enhancer in cell lines with low MGMT expression results in elevated MGMT expression. Deletion of this enhancer in cell lines with high MGMT expression leads to a dramatic reduction of MGMT and a lesser extent of Ki67 expression, increased TMZ sensitivity, and impaired proliferation. Together, these studies uncover a mechanism that regulates MGMT expression, confers TMZ resistance, and potentially regulates tumor proliferation.

[1] Biochemistry and Molecular Biology, Mayo Clinic, 200 1st St SW, Rochester, MN 55905, USA. [2] Institute for Cancer Genetics, Department of Pediatrics and Department of Genetics and Development, Irving Cancer Research Center, Columbia University, 1130 St. Nicholas Avenue, New York, NY 10032, USA. [3] Department of Neurosurgery, Tongji Hospital of Tongji Medical College, Huazhong University of Science and Technology, No.1095 Jie Fang Avenue, Hankou, 430030 Wuhan, China. [4] Department of Radiation Oncology, Mayo Clinic, 200 1st St SW, Rochester, MN 55905, USA. [5] Department of Neurologic Surgery, Mayo Clinic, 200 1st St SW, Rochester, MN 55905, USA. [6] Department of Laboratory Medicine and Pathology, Mayo Clinic, 200 1st St SW, Rochester, MN 55905, USA. Correspondence and requests for materials should be addressed to J.N.S. (email: Sarkaria.Jann@mayo.edu) or to Z.Z. (email: zz2401@cumc.columbia.edu)

Glioblastoma (GBM) is the most common and aggressive primary brain tumor. Despite current therapy, median survival for GBM patients is ~15 months[1]. Temozolomide (TMZ) has been the standard chemotherapy for newly diagnosed GBM for more than a decade[2]. However, almost all patients eventually develop resistance. Moreover, recurrent tumors in general are more aggressive than primary tumors. Therefore, there is a critical need to understand how TMZ resistance is acquired and whether there is any connection between TMZ resistance and tumor aggressiveness.

While multiple factors have been associated with TMZ resistance, expression of O6-methylguanine-DNA-methytransferase (MGMT) remains a major cause[3]. MGMT is a DNA repair protein, which removes the cytotoxic $O_6$-methylguanine ($O_6$MG) DNA lesions generated by TMZ, and high MGMT expression in cells is mechanistically linked to robust TMZ resistance. MGMT expression can be silenced by the methylation of a promoter/enhancer (P/E) region, which contains a promoter and a 59 bp cis-acting enhancer element that spans the first exon–intron boundary of *MGMT* gene[4]. Several large cohort studies indicate an association between DNA methylation of the (P/E) region and favorable outcomes of TMZ treatment[5,6]. This observation led to the hypothesis that MGMT inhibition may be a plausible strategy for sensitizing TMZ therapy in MGMT-expressed tumors[7–9]. However, combinations of TMZ with MGMT inhibitors such as $O_6$-benzylguanine ($O_6$BG), a synthetic derivative of guanine that can inhibit MGMT but was developed before the clone of the *MGMT* gene, resulted in enhanced hematologic toxicities, a reduced therapeutic window and no clinical benefit compared to TMZ alone[10,11]. Moreover, there is discordance between promoter methylation status and MGMT protein expression in GBM with wild-type *MGMT* coding sequence[12–14]. For instance, high levels of MGMT expression were detected in samples with DNA methylation at the P/E region. These observations indicate that, in addition to promoter methylation, other factors may regulate MGMT expression and confer TMZ resistance, and that identification of these additional mechanisms of MGMT regulation may provide a strong rationale for the development of a new class of drugs for this deadly disease.

Besides DNA methylation, posttranslational modifications on histone proteins also regulate gene expression. Distinct histone modifications are found at gene regulatory elements important for gene transcription[15]. For example, H3K4me3 is enriched at promoters of active genes, whereas H3K27me3 is enriched at promoters of repressed genes. In addition to promoters, enhancers, a DNA element that promotes the gene transcription via a long-range interaction with their cognate promoters, are surrounded by nucleosomes with distinct histone modifications[16]. Based on the histone modifications of surrounding nucleosomes, enhancers in general can be classified as active, primed, or poised ones. Active enhancers are typically surrounded by nucleosomes with H3K4me1 and H3K27ac[17], with the levels of H3K27ac correlating positively with enhancer activity. Primed enhancers are marked with nucleosomes with H3K4me1, whereas poised enhancers are marked by both H3K4me1 and the repressive mark H3K27me3[18]. Here, we report the identification of a distal enhancer, which we call the K-M enhancer that is situated between *MKI67* and *MGMT* promoters and is 560 kb away from the *MGMT* promoter. Ki67 is a well-known cell proliferation mark for many tumor types including GBM[19]. In other cancers, such as breast cancer[20–22], the fraction of cells staining positively for Ki67 is associated with increased proliferation and adverse clinical outcome. While controversial, high Ki67 staining is associated with elevated proliferation and poor prognosis of brain tumors in some of these studies[23–26]. We show that the K-M enhancer activates *MGMT* gene expression even in the presence

of a hypermethylated promoter. Moreover, deletion of the enhancer results in reduced expression of MGMT, but to a lesser extent on Ki67 expression. Finally, brain tumor cells lacking the enhancer are sensitive to TMZ and exhibit reduced growth rate. Together, these studies uncover a previously unknown mechanism regulating both TMZ resistance and the proliferation of GBM cells.

## Results

**Enhancer marks are altered in a TMZ-resistant GBM line**. To investigate the epigenetic changes that occur during tumor recurrence, we used a patient-derived xenograft (PDX) model previously described[14]. The GBM12 xenograft line derived from a newly diagnosed MGMT hypermethylated tumor was used to generate TMZ-resistant sublines. For this, multiple mice with flank tumors generated from GBM12 were treated with three cycles of TMZ or placebo. Two tumor sublines, a TMZ-sensitive tumor from the placebo group named GBM12 5199 (5199) and a TMZ-resistant tumor from the TMZ treatment group named GBM12 3080 (3080) were obtained (Fig. 1a). Similar to the original hypermethylated GBM12 tumor, the placebo-treated 5199 line had low MGMT protein expression and was highly susceptible to TMZ. In contrast, the TMZ-resistant 3080 line had robust MGMT expression despite of the presence of *MGMT* promoter methylation (Fig. 1b, c). These results suggest that other mechanisms are driving MGMT expression even in the presence of the *MGMT* promoter hypermethylation.

In addition to DNA methylation, histone modifications also play an important role in the regulation of gene expression[14,27]. To test whether changes in histone modifications were associated with elevated MGMT expression in 3080, we analyzed histone modifications for active enhancers (H3K4me1 and H3K27ac), promoters (H3K4me3 and H3K9ac), or gene bodies (H3K36me3) and repressed mark (H3K9me3) in 5199 and 3080 sublines using chromatin immunoprecipitation coupled with next-generation sequencing (ChIP-seq) (Fig. 1d–g, Supplementary Fig. 1 and Supplementary Table 1). We found the global levels of H3K4me3 and H3K9ac at promoters and H3K36me3 at gene bodies were similar between 5199 and 3080. Interestingly, although global H3K4me1 and H3K27ac levels were quite similar between these two sublines based on western blot analysis (Fig. 1b), their enrichment at enhancer regions was reduced in the 3080 line compared to the 5199 line, though this difference needs to be corroborated by further tests using spiked-in internal controls (Fig. 1d, e). Moreover, the level of H3K9me3 was increased at the transcription starting sites (TSS) in the TMZ-resistant 3080 compared to the TMZ-sensitive 5199 (Supplementary Fig. 1). We note a previous report showing that heterochromatin reorganizes and H3K9me2/3 increases in GBM cells treated with TMZ[28]. These results suggest that two histone modifications at enhancer regions are susceptible to alterations during the acquisition of TMZ resistance.

Activated enhancers are characterized by increased H3K27ac levels surrounding the enhancers and will lead to elevated transcription of their target genes. To classify enhancers that are altered in the 3080 compared to 5199 sublines, we performed unsupervised clustering analysis based on the changes of H3K4me1 and H3K27ac levels and the correlations with the expression change of genes close to each of these putative enhancers (Fig. 1h). In comparison to 5199, we observed that most enhancers exhibited reduced levels of H3K27ac and H3K4me1 in 3080, consistent with our previous observation that these two marks were reduced at enhancer regions in the 3080 line compared to 5199 line. A small group of enhancers (1141 Group-1 enhancers) exhibited increased H3K27ac and H3K4me1

in the 3080 line, suggesting that this group of enhancers is activated in the TMZ-resistant line. Moreover, the activation of this group of enhancers correlated with increased expression of nearby genes. Pathway analysis of nearby genes of this group of enhancers indicated that genes involved in gliomagenesis and cancer drug resistance are enriched (Fig. 1i). Interestingly, *MGMT*, a key driver of TMZ resistance[29], was one of the top 10 genes in this list that shows significant changes in gene expression (Table 1 and Supplementary Data 1). These results

suggest that a subgroup of enhancers were activated in the 3080 line, at least one of which contributes to TMZ resistance.

**MGMT is regulated by a novel enhancer**. Inspection of H3K4me1, H3K27ac, H3K4me3, and H3K36me3 ChIP-seq peaks close to the *MGMT* gene locus revealed that H3K36me3 within the *MGMT* gene body and H3K4me3 in the promoter region of *MGMT* were enriched in the 3080 line compared to 5199 line, consistent with the increased *MGMT* transcription in the 3080

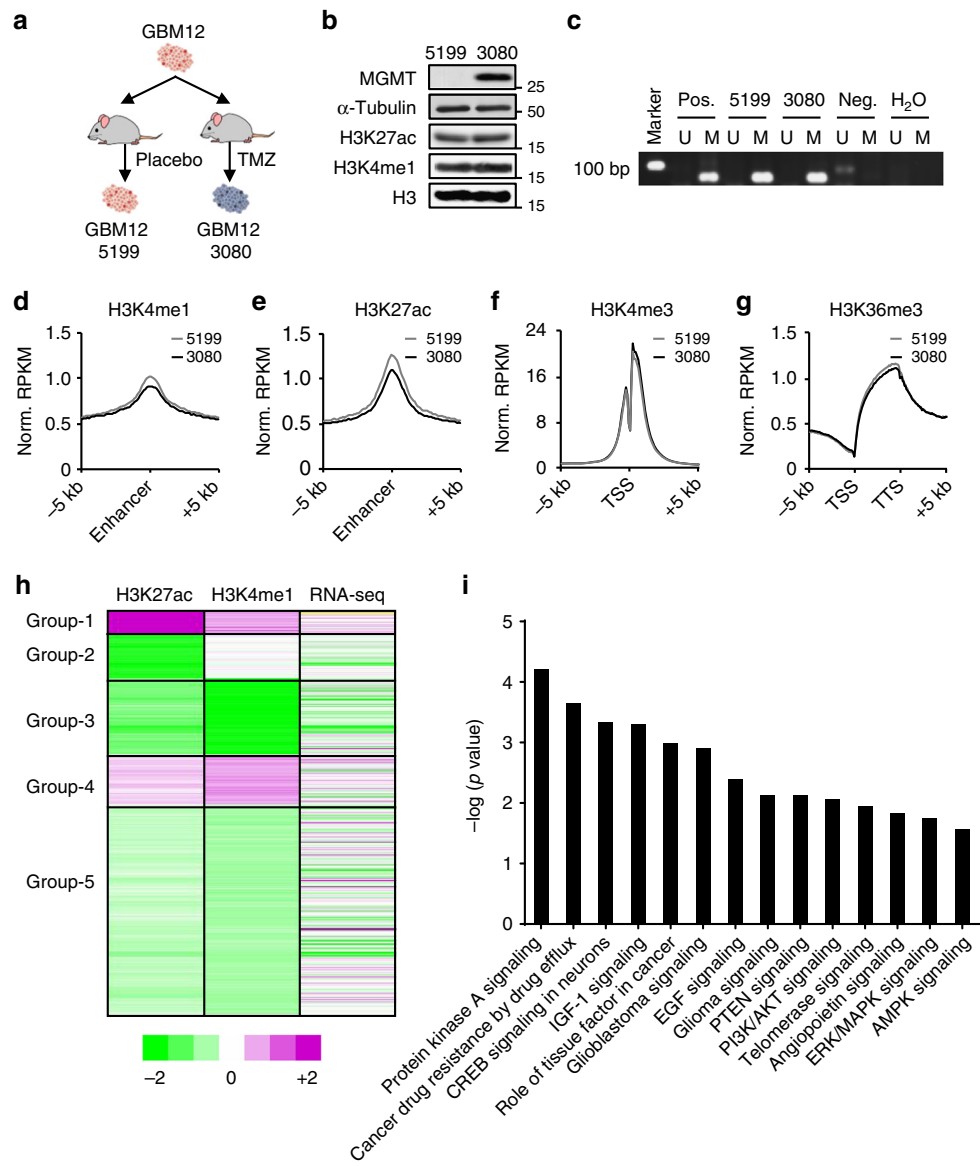

**Fig. 1** Enhancer marks are altered in a TMZ-resistant PDX line. **a** Schematic diagram for the development of 5199 and 3080 xenograft lines from parental GBM12 cells. Mice with GBM12 flank tumors were treated with placebo or three cycles of TMZ (50 mg/kg/d for 5 days every 28 days). Tumors were dissected after reaching >1500 mm³. The xenograft subline established from placebo-treated tumor was named 5199 and that from TMZ-treated tumor was named 3080. **b** Analysis of MGMT, H3K27ac and H3K4me1 levels in protein lysates of 5199 and 3080 xenograft tissues by western blotting. α-tubulin and histone H3 was used as a loading control for cytosolic proteins and nuclear proteins, respectively. **c** Analysis of MGMT promoter methylation in 5199 vs. 3080 xenograft lines by MS-PCR. Universal methylated DNA was used as positive control and normal human brain DNA was used as negative control. **d–g** Aggregate plots showing the average ChIP-seq reads distribution for the various histone marks: H3K4me1 (**d**) and H3K27ac (**e**) ChIP-seq reads surrounding the published enhancer regions; H3K4me3 (**f**) ChIP-seq reads surrounding transcription start site (TSS); H3K36me3 ChIP-seq reads across gene bodies (**g**). TTS: transcription termination site. **h** Heat maps showing cluster analysis, based on H3K27ac and H3K4me1 alterations, and expression change for genes closest to each genomic locus. The ratio of ChIP-seq reads density (3080 reads/5199 reads) for H3K4me1 and H3K27ac was calculated and subsequently analyzed by k-means cluster analysis. Changes in gene expression were calculated as RNA-seq reads ratio (3080 reads/5199 reads). Color scale represents decreased (green) and increased (magenta) signal intensity (intensity $= \log_2 \frac{3080\ \text{reads}}{5199\ \text{reads}}$). **i** Bar graphs showing ingenuity pathway analysis for Group-1 genes. Pathways with a *p* value <0.05 are presented

**Table 1 The top 10 nearby genes with the most altered Group-1 enhancers**

| Rank | Gene symbol | Associated fold change | *p* Value | Adjusted *p* value | Full gene name |
|---|---|---|---|---|---|
| 1 | MGMT | 15.63 | <0.001 | <0.001 | O-6-methylguanine-DNA methyltransferase |
| 2 | ST18 | 5.64 | 0.086 | 1 | Human suppression of tumorigenicity 18 |
| 3 | WDFY4 | 4.32 | 0.206 | 1 | WD repeat- and FYVE domain-containing protein 4 |
| 4 | C17orf50 | 4.32 | 0.894 | 1 | Chromosome 17 open reading frame 50 |
| 5 | WBSCR17 | 4.32 | 0.477 | 1 | Williams–Beuren syndrome chromosome region 17 |
| 6 | ACTL8 | 3.32 | 0.952 | 1 | Actin-like 8 |
| 7 | MAP3K21 | 3.32 | 0.604 | 1 | Mitogen-activated protein kinase kinase kinase 21 |
| 8 | SORCS3 | 3.32 | 0.775 | 1 | Sortilin-related VPS10 domain-containing receptor 3 |
| 9 | BHLHA9 | 3.32 | 1 | 1 | Basic helix-loop-helix family member A9 |
| 10 | TTR | 3.32 | 0.952 | 1 | Transthyretin |

Genes with the most elevated expression within the 1141 genes in Group-1 are listed in the table. Associated fold changes represent the fold change in gene expression in the 3080 subline over 5199 line (as calculated by normalized RNA-seq reads in 3080 line divided by normalized RNA-seq reads in 5199 line). Genes are sorted by fold change and 10 genes with the highest fold change are presented in the table. The *p* value and adjusted *p* value are also shown

line (Fig. 2a). Moreover, we detected an increase in H3K4me1 and H3K27ac enrichment in the 3080 line compared to 5199 line at a region 560 kb away from the *MGMT* promoter, suggesting that this region may be a putative enhancer that can activate MGMT expression. Because this putative enhancer localized in an intergenic region between *MKI67* gene and *MGMT* gene, we named this putative enhancer K-M enhancer. By analyzing immunoprecipitated chromatin DNA using four pairs of primers, including three pairs (PE1-3) spanning the putative K-M enhancer region and a control region 5 kb away from putative enhancer region (PE + 5 kb), with quantitative PCR (ChIP-qPCR), we confirmed that the levels of H3K27ac were significantly higher in the 3080 line than the 5199 line (Fig. 2b), whereas an equivalent level of H3K4me1 within the putative enhancer was detected between these two lines. The discrepancy in H3K4me1 at the putative enhancer detected by ChIP-qPCR and ChIP-seq may be due to the fact that H3K4me1 can be detected in the distal region. Alternatively, this putative enhancer could be in primed state, characterized by the presence of H3K4me1 and absence of H3K27ac in the 5199 line. Collectively, these results suggest that a putative enhancer is activated in the TMZ-resistant 3080 line and potentially drives MGMT expression.

Enhancers are regulatory elements that can promote gene expression. Therefore, we first tested whether this region can enhance transcription of a luciferase reporter gene. We cloned ten DNA fragments (R1–R10), each 1–2 kb in size, spanning the 13.5 kb H3K27ac peak region, in front of an SV40 promoter-driven firefly luciferase reporter. Each reporter construct was co-transfected with a pRL *Renilla* luciferase control reporter construct, which constitutively expresses *Renilla* luciferase to allow for normalization of transfection efficiency. While R1, R2, R6, R7, R9, and R10 significantly stimulated transcription compared to pGL3 vector control (Supplementary Fig. 2), only the 1.5 kb R7 fragment, localized in the second H3K27ac peak region, exhibited significantly higher activity in the 3080 line compared to 5199 (Fig. 2c). These results suggest that R1, R7, and R10 regions have enhancer activity in luciferase reporter assays, with the R7 fragment exhibiting differential effect on the reporter in the 3080 compared to 5199.

Enhancers typically contact with their cognate gene promoters through long-range interactions[30–32]. We next tested whether the putative enhancer interacts with the *MGMT* promoter using the chromatin conformation capture (3C) assay (Fig. 2d). In 3080 cells, a strong interaction between K-M enhancer and *MGMT* promoter was identified. The fourth test fragment (F4), which overlaps with the R7 region tested in the reporter assay, displayed significantly higher interaction frequency with the *MGMT* promoter compared to the neighboring DNA fragments. Moreover, the interaction frequency between the F4 fragment and the *MGMT* promoter in the 5199 line lacking MGMT expression was significantly lower than the 3080 line, supporting the idea that the putative K-M enhancer region specifically interacts with the *MGMT* promoter in the 3080 line. Therefore, the 1.5 kb region located 560 kb away from the *MGMT* promoter has characteristics of an active enhancer (surrounded by nucleosomes with high H3K27ac and H3K4me1, transcription enhancement, and interaction with promoter). The differential activity between the placebo 5199 line and the TMZ-resistant 3080 line further underscored the idea that this enhancer was only activated in the 3080 line to stimulate *MGMT* transcription, even in the presence of a methylated *MGMT* promoter.

**Enhancer activation in PDX lines and primary tumor samples.** In addition to the 3080 line, we have detected MGMT protein expression in four *MGMT* promoter hypermethylated xenograft lines previously. To investigate whether the enhancer was activated in any of these MGMT-expressing and promoter-methylated PDX lines, we analyzed H3K4me1 and H3K27ac using ChIP-qPCR in eight *MGMT* promoter-methylated PDX lines including four PDX lines expressing MGMT (GBM43, GBM64, GBM115, and GBM122) and four lines without MGMT expression (GBM46, GBM59, GBM61, and GBM102) (Supplementary Fig. 3a). H3K4me1 ChIP-qPCR analysis showed that six PDX lines (GBM43, GBM64, GBM115, GBM46, GBM61, and GBM102) had high levels of H3K4me1 at the enhancer region compared to a fragment 5 kb away (Supplementary Fig. 3b). H3K27ac ChIP-qPCR analysis indicated that of the six samples with H3K4me1, three samples (GBM64, GBM115, and GBM46) had higher levels of H3K27ac at the K-M enhancer region compared to control locus (Supplementary Fig. 3c), suggesting that the K-M enhancer is activated in these three lines. The GBM46 line is a MGMT low-expressing line from a recurrent tumor, whereas the two MGMT-expressed lines, GBM115 and GBM64, are from primary and recurrent tumor, respectively, and have high levels of MGMT expression. Although the levels of H3K4me1 and H3K27ac enrichment at the K-M enhancer locus

do not correlate with MGMT expression in GBM46 line, activation of the enhancers in two other *MGMT* promoter-methylated PDX lines correlates with high levels of MGMT expression, suggesting that enhancer activation is one explanation for the discordance between *MGMT* promoter methylation and gene expression in both primary and recurrent tumor lines.

To determine whether enhancer activation can also be detected in primary tissues, we chose to analyze H3K4me1 and H3K27ac at the enhancer locus in paired primary and recurrent GBM tumors with a methylated *MGMT* promoter. Sixty-four paired frozen patient tumor samples at Mayo Clinic were identified and further filtered based on the presence of *MGMT* promoter

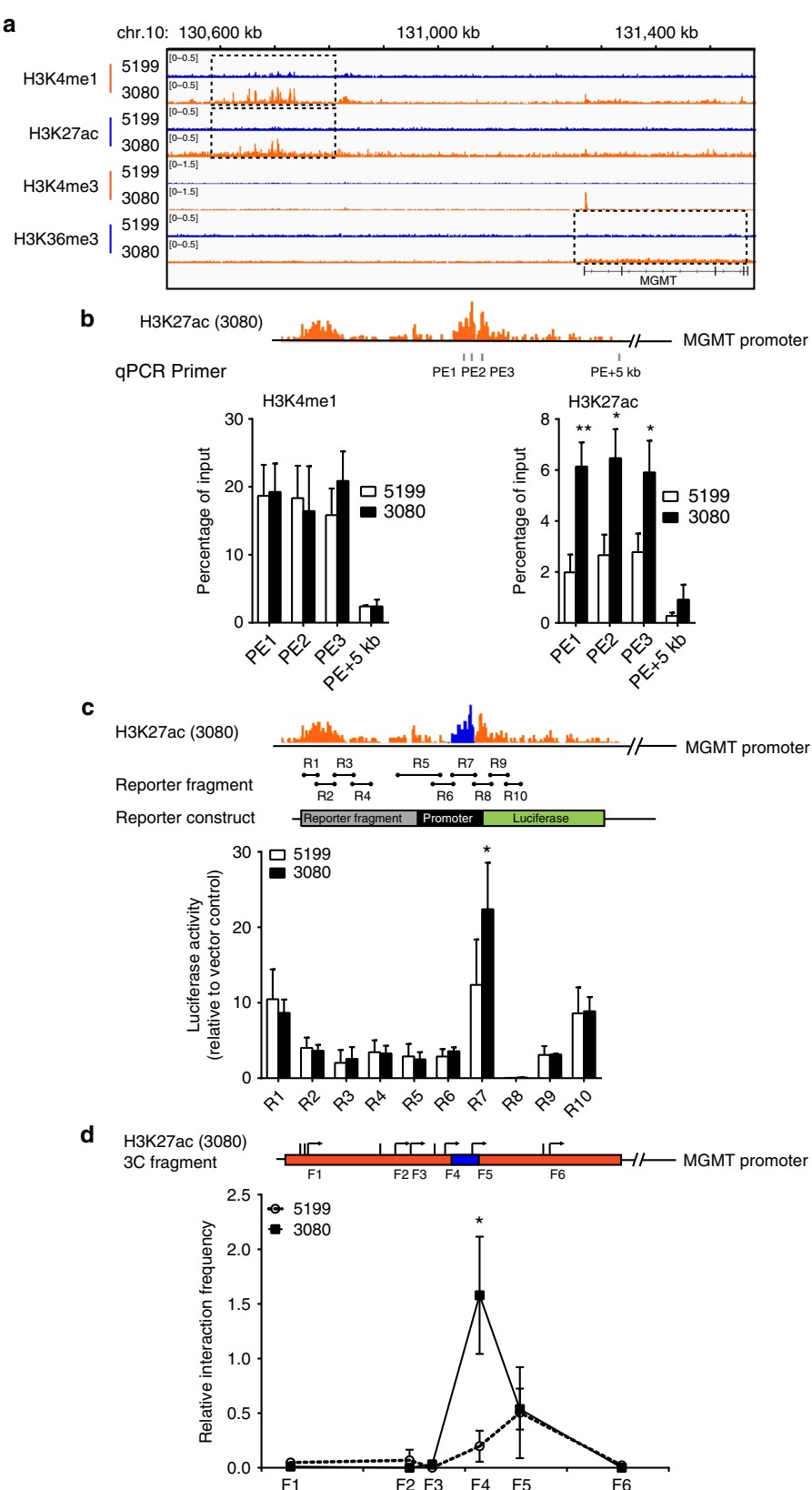

hypermethylation, TMZ treatment prior to resection of recurrent disease and the suitable tumor cell density and tissue quality to yield three pairs for subsequent ChIP-qPCR analysis (Table 2). Compared to the primary tumor, both H3K4me1 and H3K27ac levels increased in the recurrent tumor from patient #1 (Fig. 3a, b). In contrast, while H3K27ac increased slightly in the recurrent tumor from patients #2 and #3, the enrichment of H3K4me1 was not altered at their enhancer locus. Interestingly, we detected a significant increase in MGMT expression in the recurrent tumor from patient #1 compared to its matched primary tumor (Fig. 3c), whereas MGMT expression was not significantly altered in recurrent tumors from patients #2 and #3. Based on the immunofluorescence staining, we estimated that the fraction of MGMT-expressing cells increased from 4% in the primary tumor of patient #1 to 24% in recurrent tumor of this patient. A similar percentage of MGMT-expressing cells were also detected in the 3080 subline (Supplementary Fig. 4). The relatively low percentage of MGMT-positive cells detected in patient #1 and in 3080 subline is likely due to tumor heterogeneity and/or relatively low sensitivity of MGMT immunofluorescence. Supporting this idea, glioblastoma is known for its extensive intratumor heterogeneity[33]. It has been shown previously that only 1 out of 50 patient samples had over 50% MGMT-positive cells, and most MGMT-expressed tumor samples have 10–50% MGMT-positive cells[34]. The fraction of MGMT-expressing proliferating cells was not increased in recurrent tumors of patients #2 and #3 compared to their corresponding primary tumors (Fig. 3d–f and Supplementary Fig. 5). Although the small sample size precludes robust conclusions, the data suggest activation of this enhancer may also drive the MGMT expression in some recurrent tumors with MGMT promoter methylation. In the future, it would be interesting to analyze the enhancer activity in a larger cohort of patient samples at single cell levels.

**Targeting p300 to the K-M enhancer increases MGMT expression.** The level of H3K27ac at the enhancers correlates with enhancer activity and gene expression. Therefore, we tested whether an increase in H3K27ac on nucleosomes surrounding the enhancer element in MGMT low-expressing cells will affect MGMT expression[35]. Briefly, a Flag-tagged nuclease deactivated Cas9 (dCas9) protein was fused with p300 HAT domain[36] and was targeted to the enhancer locus using five-guide RNAs (gRNAs) in cells with low levels of MGMT expression (Fig. 4a). HEK293T cells were chosen first in this experiment due to their high transfection efficiency. Successful targeting of dCas9 or dCas9$^{\text{p300 Core}}$ fusion proteins to the K-M enhancer locus was confirmed by Flag ChIP-qPCR (Fig. 4b). Moreover, we observed a marked increase of H3K27ac at the K-M enhancer locus when dCas9$^{\text{p300 Core}}$ fusion protein, but not dCas9 alone, was expressed along with gRNAs (Fig. 4c). RT-PCR analysis indicated MGMT transcription also increased dramatically in HEK293T cells when

targeting dCas9$^{\text{p300 Core}}$ to the enhancer region by CRISPR/dCas9 (Fig. 4d). Although expression of the dCas9$^{\text{p300 Core}}$ fusion protein alone without gRNAs slightly increased H3K27ac levels, this increase did not significantly alter MGMT expression, suggesting that a threshold level of H3K27ac is needed to alter the chromatin state surrounding the enhancer and gene expression of MGMT. We also observed a slight, but significant increase in MGMT expression after we targeted dCas9$^{\text{p300 Core}}$ to the enhancer locus in the 5199 line (Fig. 4e–g). Thus, the increased H3K27ac at nucleosomes surrounding the K-M enhancer can stimulate MGMT expression even in the presence of MGMT promoter methylation.

**Deletion of the enhancer results in reduced MGMT expression.** To directly determine whether the enhancer was essential for MGMT expression, we used the CRISPR/Cas9 system[37] to delete a 3.3 kb region (chr10: 130,704,894–130,708,206) in the MGMT-expressing SKMG3 glioblastoma cell line (Fig. 5a). Four independent clones (three homozygous deletion clones and one wild-type control clone) were isolated. The genotypes of those four clones were identified by PCR and further confirmed by Sanger sequencing (Supplementary Fig. 6a–b). Compared to parental SKMG3 cells and wild-type control clone, MGMT transcription and protein expression level were markedly reduced in enhancer-deleted clones (Fig. 5b, c).

Next, we analyzed the TMZ sensitivity of SKMG3 clones using the clonogenic assay. Compared to the parental SKMG3 line and wild-type control clone, the enhancer-deleted clones were significantly more sensitive to TMZ treatment, with a log-fold decrease in IC$_{50}$ from over 100 μM to less than 10 μM (Fig. 5d–f). The sensitizing effect was not observed in the group pre-treated with the MGMT inhibitor O$_6$BG (Fig. 5e, f), indicating the change in TMZ sensitivity was mechanistically linked to MGMT activity. Since this TMZ concentration range is clinically achievable, these results suggest that an efficient suppression of the enhancer activity could be a strategy to overcome the emergence of TMZ resistance.

To further narrow down the location of the enhancer-activating MGMT expression in SKMG3 cells, we designed an additional guide RNA (g3) targeted at the middle of the other two-guide RNAs used for the 3.3 kb deletion and generated two different 1.5 kb deletions when combined with g1 or g2 guide RNAs, respectively (Supplementary Fig. 7a). Depletion of the first 1.5 kb region (ch10: 130,704,894–130,706,549), named Del1 region, also resulted in a dramatic reduction in MGMT suppression in SKMG3 cells (Fig. 5g), while deletion of the Del2 region, the 1.8 kb region closer to the MGMT promoter, did not cause a significant change in MGMT expression (Supplementary Fig. 7b). This result further narrowed down the location of K-M enhancer to a 1.5 kb region that lies about 560 kb away from the MGMT promoter and is essential for MGMT expression. Similar to larger deletion clones, SKMG3 cells with

**Fig. 2** Delineation of a putative enhancer-associated with *MGMT* gene in 3080 subline. **a** IGV snapshots showing H3K4me1, H3K27ac, H3K4me3, and H3K36me3 ChIP-seq reads density at the putative enhancer region, *MGMT* promoter, and gene body in 5199 (TMZ-sensitive) and 3080 (TMZ-resistant) xenograft lines. **b** H3K4me1 (bottom left) and H3K27ac (bottom right) occupancy at the putative K-M enhancer in 5199 (parental) and 3080 (TMZ-resistant) xenograft tumors were analyzed by ChIP-qPCR. Three different sets of primers (PE1-3) were designed to analyze against putative enhancer region. A set of primers amplifying 5 kb away from the putative enhancer (PE + 5 kb) was used as negative control. Top: magnified view of H3K27ac ChIP-read peaks and the location of each primer set. **c** Effect of each fragment on transcription of luciferase reporter gene in 5199 and 3080 cells. Fragments R1–R10 covering putative enhancer region (top) were cloned upstream of luciferase promoter (middle). The ChIP-seq peak covered by R7 region is labeled in blue. Each construct was transfected into 5199 or 3080 cells along with a plasmid expressing *Renilla* luciferase. Firefly and *Renilla* luciferase activities were measured 36 h after transfection. Luciferase activity was normalized to the *Renilla* luciferase activity (internal control) first and subsequently normalized to an empty vector control. **d** Enhancer–promoter interaction analyzed by chromatin conformation capture (3C) assay. Relative interaction frequency of each restriction fragment (F1–F6) was calculated as described in the experimental procedures and was plotted on genomic location of the 3′ cutting site of each fragment (X axis). Values reported were derived from three biological repeats (*$p < 0.05$, **$p < 0.01$, Student's *t* test). The genomic region covered by H3K27ac peak is labeled in orange, location of R7 fragment was colored in blue to highlight the overlap with the F4 fragment

**Table 2 Treatment and survival information for patients who provided primary and recurrent tumor samples**

| Demographics | Patient 1 | Patient 2 | Patient 3 |
|---|---|---|---|
| Age | 64 | 67 | 68 |
| Treatment | Standard chemo-radiotherapy | Standard chemo-radiotherapy | Standard chemo-radiotherapy |
| Initial progression-free survival | 11 months | 21 months | 23 months |
| Initial treatment at progression | TMZ | TMZ | TMZ |
| Interval between primary and recurrent tumor resection | 17 months | 29 months | 38 months |
| Overall survival | 29 months | 55 months | 43 months |

The patient demographics and treatment records were summarized in this table

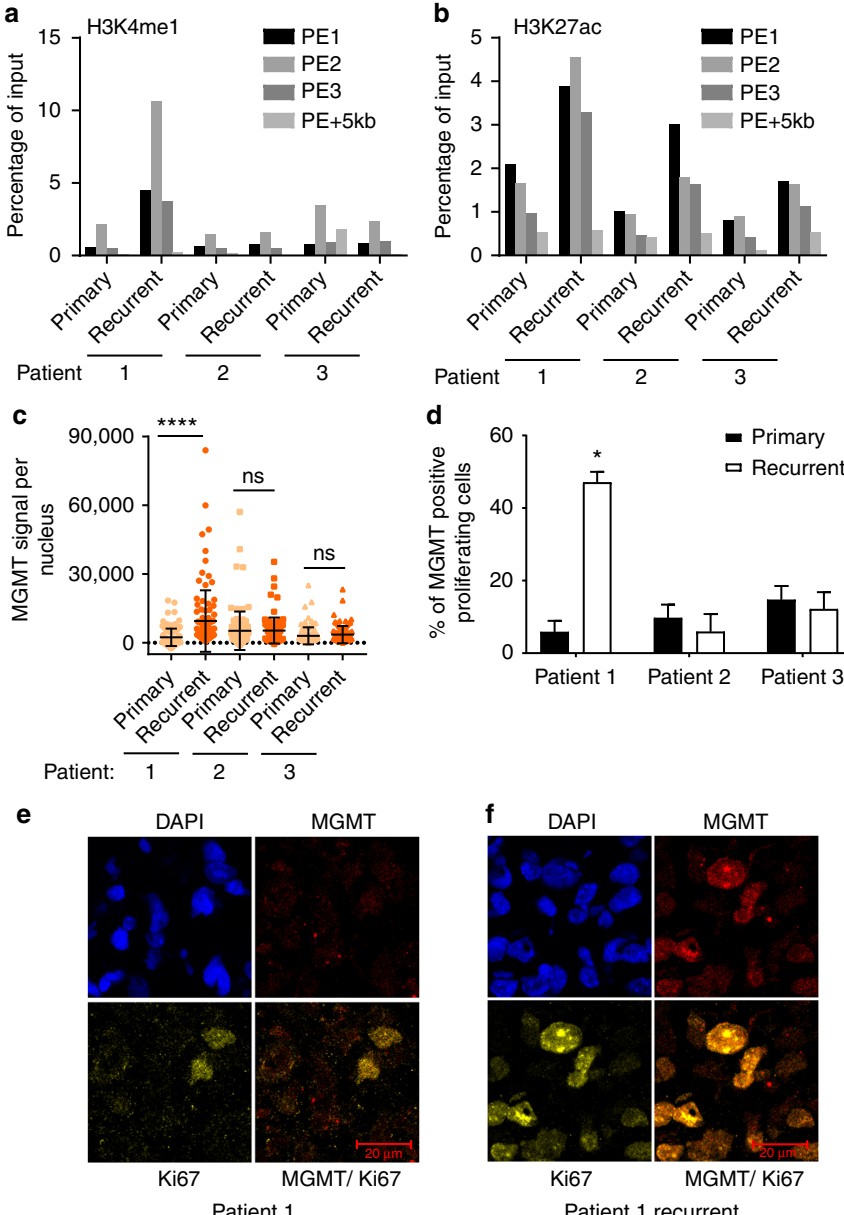

**Fig. 3** Enhancer activity and MGMT expression in three pairs of GBM patient samples. **a**, **b** Relative occupancy of H3K4me1 (**a**) and H3K27ac (**b**) in the K-M enhancer region in primary and matched recurrent tumors from three GBM patients was analyzed by ChIP-qPCR using primers described in Fig. 2. **c** MGMT expression level in each tumor was analyzed by immunofluorescence. The MGMT signal (area × average intensity) from 100 individual nuclei was measured and one-way ANOVA was used for the calculation of significance (****$p < 0.0001$, ns not significant, $n = 100$) **d** Analysis of MGMT expression in proliferating tumor cells. Multicolor immunofluorescence was performed using antibodies against CD45, Ki67, and MGMT. Proliferating tumor cells (Ki67 staining without CD45 staining) were identified and the percentage of MGMT-positive proliferating cells was calculated. Two-tail paired Student's $t$ test was used for calculation of significance (*$p < 0.05$). **e**, **f** Representative images showing co-localization of MGMT and Ki67 in tissue sections from primary tumor (**e**) and recurrent tumor (**f**) obtained from patient #1

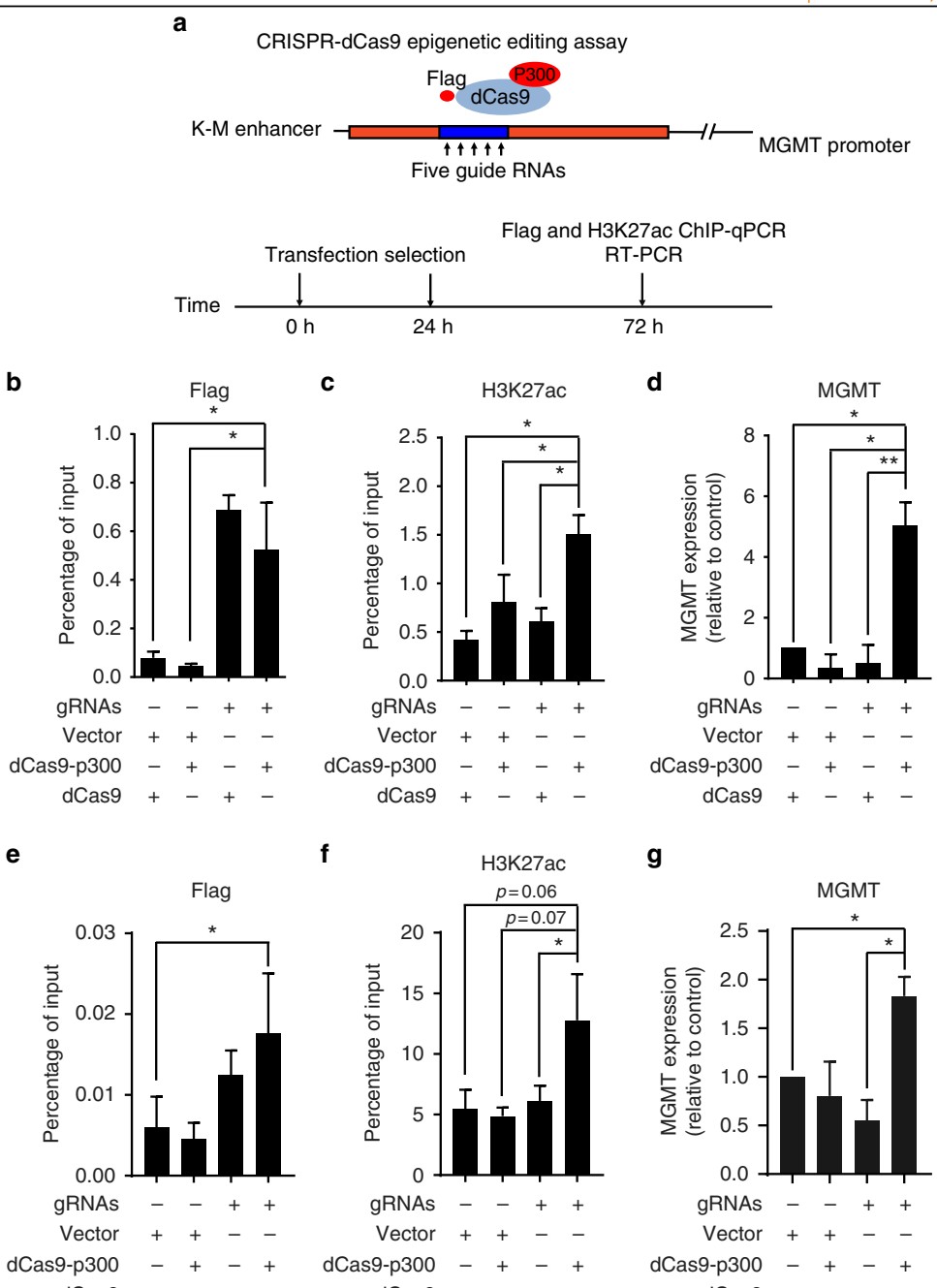

**Fig. 4** Targeting p300 to the K-M enhancer increases *MGMT* gene transcription. **a** Schematic diagram showing procedure of targeting p300 catalytic domain (p300 core) by CRISPR/dCas9 system. R7 region identified by previous experiment was labeled as blue to indicate the targeting loci. Experimental design involved co-transfection of dCas9$^{p300\ Core}$ (or control dCas9) vectors along with five-guide RNAs (or a control empty vector) followed by brief selection of transfected cells under 5 μg/ml puromycin and subsequently analyzed by ChIP-qPCR. **b, c** Analysis of the occupancy of Flag-tagged dCas9/ dCas9$^{p300\ Core}$ protein (**b**) or histone H3K27ac (**c**) at the K-M enhancer region by ChIP-qPCR in HEK293T cells 72 h after transfection. An empty vector (lentiGuide puro vector) or a mixture of guide RNA constructs were transfected along with a dCas9 or dCas9$^{p300\ core}$ expression vector. **d** MGMT expression in HEK293T cells was analyzed 72 h after co-transfection with either an empty vector or guide RNA construct along with a dCas9 or dCas9$^{p300}$ $^{core}$ expression vector. *MGMT* transcript level was first normalized to actin and subsequently calculated as fold change relative to double-negative (Vector dCas9) control (*$p < 0.05$, $n = 3$ independent experiments, Student's *t* test). **e, f** Analysis of the occupancy of Flag-tagged dCas9/dCas9$^{p300\ core}$ protein (**e**) or histone H3K27ac (**f**) at the K-M enhancer region by ChIP-qPCR in 5199 cells 72 h after infection. 5199 cells were infected with a mixture of virus containing an empty vector (lentiGuide puro vector) or five-guide RNA constructs and transfected with a dCas9 or dCas9$^{p300\ Core}$ construct. **g** The *MGMT* transcript level in 5199 cells after targeting p300 core to the enhancer by CRISPR/dCas9. The experiments were performed as described from **b-d** (*$p < 0.05$, $n = 3$ independent experiments, Student's *t* test)

the 1.5 kb deletion also showed increased TMZ sensitivity, but to lesser degree than deletion clones with the 3.3 kb region (Fig. 5h), suggesting that the second half of the 3.3 kb region also contributes to the regulation of MGMT expression. Finally, we obtained one clone with the 1.5 kb Del1 in 3080 line

(Supplementary Fig. 7e), and deletion of this fragment in the 3080 cells also resulted in reduced MGMT expression (Fig. 5i) and increased TMZ sensitivity in vitro (Fig. 5j). Together, these results fully demonstrate that this fragment can serve as the MGMT enhancer.

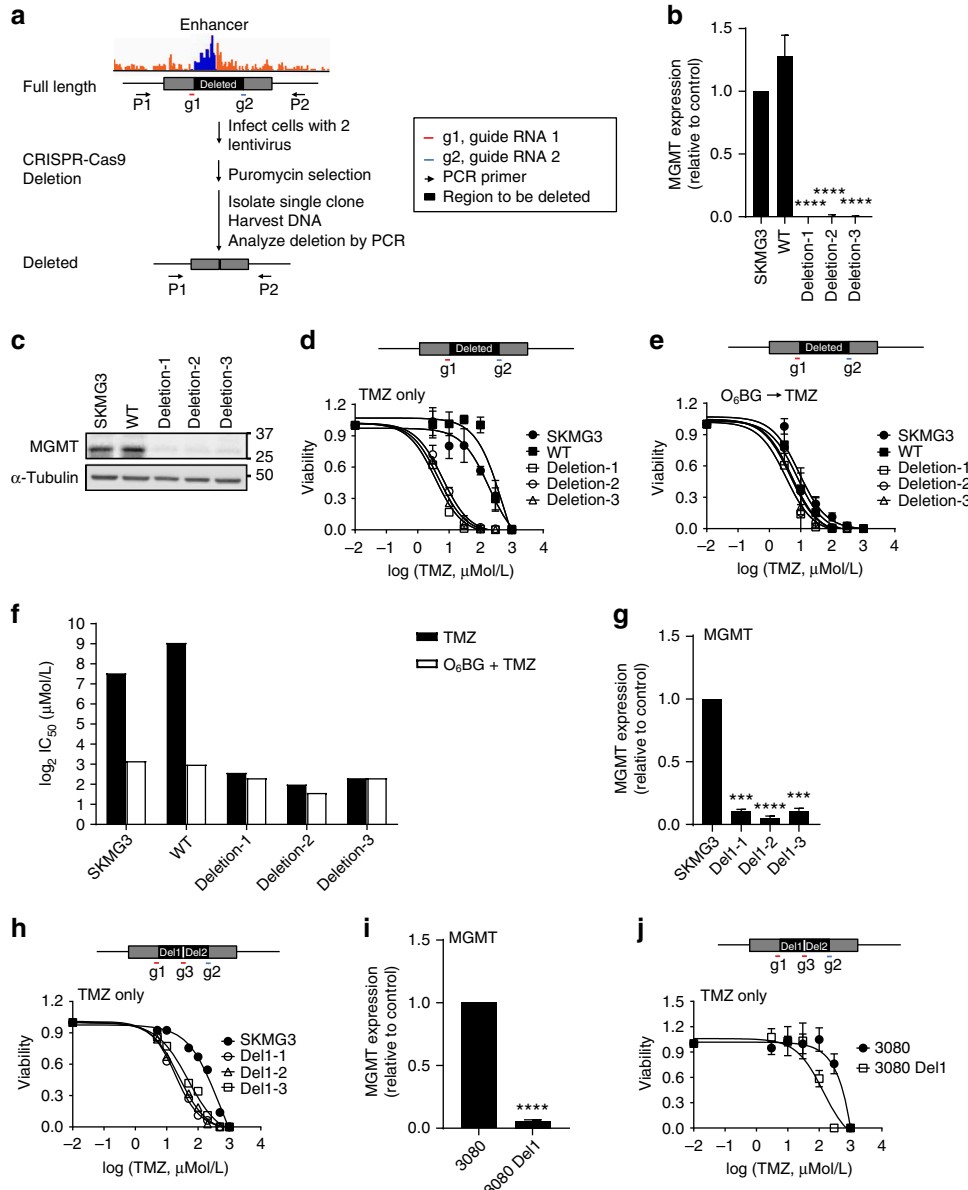

**Fig. 5** Enhancer deletion reduces MGMT expression and increases sensitivity to TMZ. **a** An outline of deletion strategy using CRISPR/Cas9 system. Cells were infected with a mixture of two virus containing guide RNAs surrounding K-M enhancer locus. After puromycin selection, single clones were isolated and tested by PCR. **b** MGMT expression in SKMG3 parental cell, SKMG3 wild-type clone, and three K-M enhancer-deleted clones were analyzed by quantitative RT-PCR. MGMT transcript level was first normalized to actin and subsequently calculated as fold change relative to SKMG3 parental line. **c** MGMT protein levels in SKMG3 parental cell, SKMG3 wild-type clone, and three K-M enhancer-deleted clones were tested by western blotting. α-Tubulin was used as loading control. **d** SKMG3 parental line, wild-type clone, and K-M enhancer-deleted clones were treated with indicated concentrations of temozolomide (0–1000 μM/L final concentration). Cell viability was determined using clonogenic assay. **e** SKMG3 parental line, wild-type clone, and K-M enhancer-deleted clones were treated with 10 μM O6BG 1 h prior to temozolomide (0–1000 μM/L final concentration). Cell viability was determined using clonogenic assay. **f** Bar graph showing $\log_2$(TMZ $IC_{50}$) in SKMG3 clones with or without O6BG pre-treatment. **g** MGMT expression in SKMG3 parental cells and three Del1 deleted clones were analyzed by quantitative RT-PCR. *MGMT* transcript level was first normalized to actin and subsequently calculated as fold change relative to SKMG3 parental line. **h** SKMG3 parental cells and three Del1 deleted clones were treated with indicated concentrations of temozolomide (0–1000 μM/L final concentration). Cell viability was analyzed using clonogenic assay. **i** *MGMT* transcription levels in 3080 parental cells and 3080 Del1 deleted cells were analyzed by quantitative RT-PCR. *MGMT* transcript level was first normalized to actin and subsequently calculated as fold change relative to 3080 parental line **j** 3080 parental cells and 3080 Del1 deleted cells were treated with different concentration of temozolomide (0–1000 μM/L final concentration). Cell viability was determined using neurosphere formation assay ($^{***}p < 0.001$, $^{****}p < 0.0001$, $n = 3$ independent experiments, Student's *t* test)

**Effect of the enhancer deletion on Ki67 expression**. K-M enhancer may not only regulate protein expression but also promotes proliferation. During clonogenic assay analysis, the untreated deletion clones formed smaller colonies suggestive of impaired proliferation. This observation was confirmed by a proliferation assay, showing a clear decrease in proliferation rate in two out of three 3.3 kb deletion and all 1.5 kb deletion SKMG3 clones (Fig. 6a–d and Supplementary Fig. 7c–d). However, cell

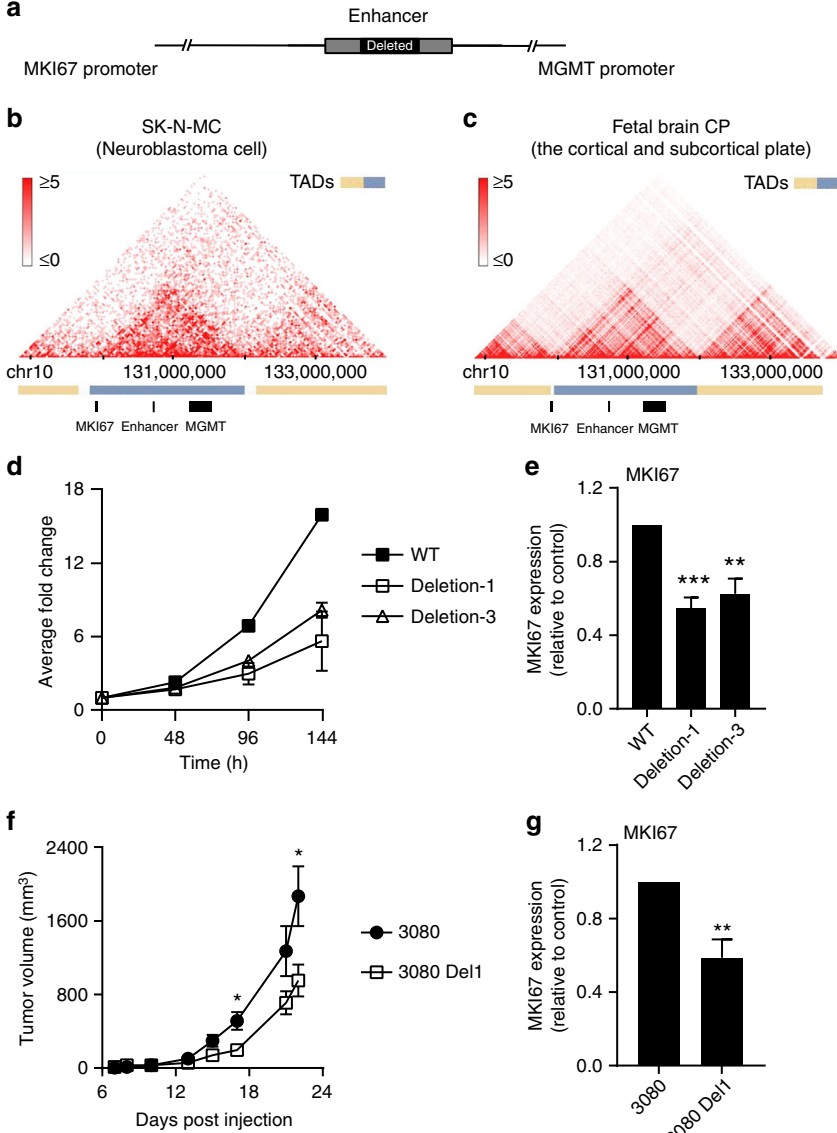

**Fig. 6** Deletion of the K-M enhancer affects cell proliferation and Ki67 expression. **a** Schematic diagram showing the relative genomic location of *MKI67* gene, the K-M enhancer, and *MGMT* gene. **b**, **c** The Hi-C analysis from SK-N-MC cells (**b**) and human fetal brain cortical plate (**c**) are taken from the 3D genome browser site[38]. The boundaries of the topologically associated domains (TADs) and the location of *MGMT*, *MKI67*, and the K-M enhancer is indicated below. **d** Cell proliferation rate was analyzed by Incucyte. The cell confluency read out for each cell was normalized by the cell confluency acquired at the first time point. **e** *MKI67* transcription levels in SKMG3 wild-type clone and two enhancer-deleted clones were analyzed by quantitative RT-PCR. Ki67 expression level was first normalized against actin and subsequently calculated as fold change relative to wild-type clone. **f** Growth rate of 3080 parental and 3080 Del1 cells injected flank tumors. Equal numbers of 3080 parental cells and 3080 Del1 cells were injected into the flank of mice (*n* = 5/ group). Tumor volume was used to represent growth rate. **g** *MKI67* transcription levels in 3080 parental cells and 3080 Del1 deleted clone cells were analyzed by quantitative RT-PCR. Ki67 expression level was first normalized against actin and subsequently calculated as fold change relative to the parental line

cycle appears to be normal in the enhancer-deleted clones. This suggests that the enhancer may also regulate expression of genes involved in proliferation in addition to MGMT.

*MKI67* is a gene encoding the nuclear protein Ki67, which is a proliferation marker for many tumors including GBM (Supplementary Fig. 8). Because *MKI67* is another gene localized close to the enhancer, we first analyzed the published Hi-C data sets[38] to determine whether the *MKI67* gene localized in the same topologically associating domain (TAD) with *MGMT*. We found that *MIK67* resided in the same TAD as *MGMT* and the enhancer in neuroblastoma cells (Fig. 6b) and localized at the left boundary of the TAD domain-containing

*MGMT* in cortical and subcortical plate cells (Fig. 6c). Since the TAD boundaries are not well defined, these results suggest that the K-M enhancer, *MKI67* and *MGMT* may be in the same TAD. We then analyzed Ki67 expression in the enhancer deletion clones. We observed that two SKMG3 clones with the 3.3 kb deletion exhibited a significant decrease of Ki67 expression as well as reduced proliferation (Fig. 6d, e). Tumor growth rate and Ki67 expression were also reduced in the 3080 Del1 deleted clone (Fig. 6f, g). However, we noticed that in all these clones with the deletion of the enhancer element, the expression of MGMT was affected more than Ki67. Collectively, our results indicate that the K-M enhancer is the enhancer for

both MGMT and Ki67, which in turn regulates TMZ sensitivity and possibly cell proliferation.

## Discussion

Previous studies have associated *MGMT* hypermethylation with gene silencing[39]. However, discordance between promoter hypermethylation and MGMT suppression is observed both in vitro and in vivo[14,40]. Here we identify a distal enhancer (K-M enhancer) regulating MGMT expression. Surprisingly, the activated enhancer appears to overcome the promoter hypermethylation and drives the MGMT expression. Therefore, the mechanism revealed in this study might contribute to the reported discordance between *MGMT* promoter methylation and expression in at least a fraction of tumors. Deletion of the K-M enhancer reduces MGMT and Ki67 expression, decrease cell proliferation, and sensitizes cells to TMZ to a clinical relevant level, which suggests potential therapeutic benefits of targeting enhancer activity. Beyond *MGMT*, a set of 1141 putative enhancers were activated in the TMZ-resistant 3080 line (Supplementary Data 2), which raises the possibility that multiple enhancers are altered in response to TMZ therapy, some of which contribute to the emergence of drug resistance.

MGMT expression is mechanistically linked to TMZ resistance, and the discordance between promoter methylation and protein expression observed in a subset of patients limits the prognostic accuracy of methylation assessment. Despite the overt requirement for protein expression of MGMT to repair TMZ-induced damage, *MGMT* promoter methylation is a more accurate predictor of TMZ resistance as compared to either RNA or protein expression. In this study, we demonstrate that activation of the K-M enhancer can drive MGMT expression despite promoter methylation. Moreover, a subset of *MGMT* hypermethylated GBM PDX lines with K-M enhancer activation expresses basal MGMT protein in association with de novo TMZ resistance. These observations may help explain why approximately a quarter of newly diagnosed *MGMT* hypermethylated GBM patient progress within the first few months of TMZ therapy[41]. Finally, the activation of the K-M enhancer during resistance emergence without corresponding changes to *MGMT* promoter methylation may partially explain the poor prognostic performance of methylation status in recurrent GBM. Mechanisms of inherent and acquired TMZ resistance extend beyond MGMT regulation. Therefore, we speculate that in addition to genetic mutations, enhancer alterations through changes in chromatin states likely contribute both to intrinsic and acquired TMZ resistance.

Beyond enhanced prognostic accuracy, therapeutic suppression of the K-M enhancer could be used to delay the emergence of TMZ resistance and/or sensitize-resistant patients to TMZ. We showed that epigenetic activation of the K-M enhancer drives TMZ resistance, while enhancer deletion results in greater TMZ sensitivity. Therefore, blocking enhancer activation may prevent enhancer-driven TMZ resistance. A similar concept has been tested in breast and lung cancer, where epigenetic inhibitors successfully prevented chemoresistance emergence driven by epigenetic alterations of gene promoters[42,43]. If successful, enhancer inhibition may prevent or delay the emergence of TMZ resistance and produce more durable responses for GBM patients, which may make a critical improvement in survival. Furthermore, deletion of the enhancer reduces proliferation and sensitizes cells to TMZ. This indicates that blocking the K-M enhancer activity potentially not only enhances TMZ response, but also reduces aggressiveness of otherwise TMZ-resistant tumors. In principle, enhancer activity can be blocked by several methods including blocking the recognition of acetylated histones and inhibition of

histone acetyltransferase activity. For example, both bromodomain inhibitors (BETi), which blocks H3K27ac recognition by bromodomain-containing proteins, and histone acetyltransferases inhibitors (HATi), which reduce enhancer activity by inhibiting H3K27 acetylation, can effectively inhibit enhancer-driven transcription activation with acceptable toxicity[44–46]. However, no study has been done to combine BETi or HATi with TMZ treatment. Similarly, a recent study demonstrated that the CDK7/12 inhibitor THZ1 prevents active enhancer formation at genes, which inhibit the emergence of chemoresistance in multiple cancer lines[47]. CDK7 and CDK12 are two genes that regulate Pol II-mediated gene transcription. Therefore, we propose that combination therapy with TMZ and a BETi, HATi, or CDK7/12i, may delay the emergence of TMZ resistance in responsive tumors and potentially sensitize drug-resistant tumors to TMZ.

In contrast to BET and HAT inhibitors, histone deacetylase inhibitors (HDACi), which globally increase histone acetylation including H3K27ac, may promote the emergence of TMZ resistance by activating the K-M enhancer. Supporting this idea, we observed that combination treatment of SAHA, an FDA-approved HDACi, with TMZ specifically promotes elevation of MGMT expression as a mechanism of TMZ resistance[14]. Our enhancer activation model suggests that acetylation of both the *MGMT* promoter and the K-M enhancer significantly contributes to this effect. Thus, any future designs of treatment strategies that combine TMZ and histone deacetylase inhibitors should be approached with appropriate caution.

Beyond mediating MGMT expression and TMZ sensitivity, enhancer deletion was associated with reduced proliferation in five out of six enhancer-deleted clones, suggesting that K-M enhancer inactivation may also reduce tumor growth. Interestingly, unlike the observed increase in MGMT expression, Ki67 expression in 5199 was similar to 3080 line, indicating differential influence of the K-M enhancer on the two most proximal genes, *MGMT* and *MKI67*. Indeed, MGMT expression was affected more dramatically than Ki67 in each of the enhancer deletion clones we tested. One likely explanation is that this enhancer is a key regulator of MGMT expression in all cells, while K-M enhancer modulation of the *MKI67* promoter is cell type specific. Potentially, sequence differences within the K-M enhancer region may differentially affect binding of transcriptional activators and/or repressors that result in altered enhancer activity within different cells. Future studies are needed to catalog the transcription factors that bind to the K-M enhancer and more fully characterize sequence variations within this region. Regardless of mechanism, the dual regulation of both MGMT and Ki67 by the K-M enhancer highlights the potential for this enhancer to influence both TMZ efficacy and tumor proliferation.

In summary, our study reveals a previously undocumented enhancer that, when activated, promotes MGMT expression and TMZ resistance in different cell lines, even in the presence of promoter methylation. This enhancer likely also regulates the expression of Ki67 as well as tumor proliferation. These findings suggest that inhibition of enhancer activity is a plausible strategy for the development of therapies and that an assessment of enhancer activation states might be useful as a predictive biomarker.

## Methods

**Cell culture**. GBM cell line SKMG3 (provided by Dr. David James) and HEK293T cell line procured from ATCC were maintained in DMEM (CORNING, 10-013-CV) supplemented with 10% fetal bovine serum (Millipore Sigma, TMS-013-B) and 1% penicillin–streptomycin (CORNING, 30-001-CI).

GBM xenograft sublines GBM12 5199 and GBM12 3080 previously generated by our lab were used to isolate primary cells. Primary cells from xenograft tissues were cultured in StemPro NSC media and supplements (ThermoFisher, A1050901) according to the manufacturer's instructions.

All cells are tested for mycoplasma contamination each month. All cells used in this paper are mycoplasma free.

**Xenograft tumors.** Frozen tumor tissues from xenografts established from primary GBMs (GBM43, GBM59, GBM61, GBM115, and GBM122) and those from recurrent GBMs (G46, G64, and G102) were previously generated by our lab[48].

**Paired patient samples.** Studies involving patient samples were approved by the Mayo Clinic Institutional Review Board (IRB number 09-003015). Paired (primary and recurrent) GBM frozen tissue samples collected from consented patients, who had hypermethylated *MGMT* promoter at primary diagnoses and received standard TMZ-based treatment, were obtained from Mayo Clinic Neuro-Oncology tissue bank. A total of three pairs of frozen tumors with high-tumor cellularity (>80%), large sample size (>30 mg), and hypermethylated *MGMT* promoter were selected. Tissues were cryosectioned and subjected to parallel immunofluorescence and ChIP-qPCR analyses. Tumor samples for ChIP-qPCR were collected by scraping the tumor dense areas from the tissue sections and used for ChIP assays as described in references [49–51].

**Antibodies.** Antibodies against MGMT (AF3794,1:1000) and α-tubulin (12G10, 1:1000) were used in western blotting analysis.

Antibodies against H3K4me1 (Abcam, Ab8895), H3K27ac (Abcam, Ab4729), H3K9ac (Abcam, Ab4441), H3K4me3 (Abcam, Ab8580), H3K9me3 (Active Motif, 39161), H3K36me3 (Active Motif, Cat #61101), and Flag (Sigma-Aldrich, 11583816001) were used for chromatin immunoprecipitation assay.

Monoclonal antibodies against CD45 (13917,1:100), MGMT (MAB16200, 1:100), Ki67 (14-5698-82, 1:100), and conjugated Alexa Fluor 488 labeled goat anti-rabbit IgG (111-545-144, 1:200), Alexa Fluor 594 labeled goat anti-mouse IgG (A11032, 1:200) and Cy5 labeled goat anti-rat IgG (112-175-167, 1:200) were used for immunofluorescence staining.

**Western blot assay.** Protein samples are separated using gel electrophoresis and transferred onto a nitrocellulose membrane. The membrane is blocked with 5% milk, probed by primary antibodies (1:1000), and subsequently probed by corresponding secondary antibody (1:5000).

**RNA isolation, reverse transcription, and real-time PCR.** RNA was extracted with RNeasy Plus kit (Qiagen, #74134) according to the manufacturer's instructions. Reverse transcription of mRNA was performed using SuperScript™ III Reverse Transcriptase (Invitrogen, 18080-085). For real-time PCR analysis, 1 μl of cDNA (25 ng of starting RNA) was amplified per reaction using the iTaq Universal SYBR Green Supermix (Bio-Rad, 172–5124) and the Bio-Rad CFX qPCR system. Primers for real-time qPCR analysis were listed in supplemental table (Supplementary Table 2).

**RNA-seq.** Total RNA was extracted as described above. The RNA quality was further evaluated using the Agilent 2100 Bioanalyzer (Agilent, Santa Clara, CA). The Illumina TrueSeq RNA Sample preparation Kit v.4.1 (Illumina Inc., San Diego, CA) was used to prepare cDNA libraries from 2 μg of total RNA for RNA-seq. Individual barcoded libraries were analyzed using Agilent 2100 Bioanalyzer (Agilent technologies). Sequencing was carried out on an Illumina HiSeq 2000 machine (Illumina) at Mayo Clinic Medical Genomic Facility.

**MS-PCR.** DNA was extracted from frozen tissues or cells using Blood & Cell Culture DNA Mini Kit (Qiagen, #13323). Isolated genomic DNA was bisulfite treated with EZ DNA methylation Gold kit (Zymo Research, #11-335B). The modified DNA was quantified by PCR. One pair of primer (MS-M-F/R) was used to PCR-methylated *MGMT* promoter. Another pair of primer (MS-U-F/R) was used to PCR unmethylated *MGMT* promoter. The sequences for PCR primers were listed in supplemental table (Supplementary Table 2).

**Chromatin immunoprecipitation.** A total of $1 \times 10^6$ cells or 15 mg homogenized frozen tissues were fixed with 1% paraformaldehyde at room temperature for 10 min, quenched with 0.125 M glycine, and lysed for 10 min on ice. The lysate was digested with MNase (NEB, Cat#M0247S) at 2000 gel unit/ml final concentration at 37 °C for 20 min and sonicated 15 cycles (30 s on, 30 s off) under high power using a Diagenode Bioruptor. Crosslinked DNA was immunoprecipitated with 2 μg antibody at 4 °C overnight, pulled down by protein G beads, washed, reverse crosslinked and purified for qPCR, and high-throughput sequencing analysis.

For ChIP-qPCR analysis, both input and immunoprecipitated DNA were quantified by real-time PCR with primers listed in supplemental table (Supplementary Table 2). DNA quantity for each ChIP sample was normalized against input DNA.

For ChIP-seq samples, after DNA purification ChIP-seq DNA libraries were prepared with the Ovation Ultralow DR Multiplex system (NuGEN). The DNA libraries were sequenced using the 51 bp paired-end sequencing method by an Illumina Hi-seq 2000.

**ChIP-seq analysis.** Raw reads from Illumina Hi-seq 2000 were aligned to the human genome (hg19) using Bowtie2 software with default parameters. Only uniquely mapped reads were used for secondary analysis. The ChIP-seq peaks were identified by MACS2 using the default calling parameter. Cutoff values for the $p$ value was set to 0.001. Genome-wide read coverage was calculated by BEDTools and visualized using Integrative Genomics Viewer. The reads density scan was performed by in-house Perl programs using the traditional normalization method: reads per kilobase per million mapped reads (RPKM).

For cluster analysis, first genomic regions with high H3K4me1 and low H3K4me3 occupancy in 5199 and 3080 cells were selected. H3K4me1 peaks were merged if their distance is less than 500 bp. The $\log_2 \frac{\text{H3K4me1(3080)}}{\text{H3K4me1(5199)}}$ and $\log_2 \frac{\text{H3K27ac(3080)}}{\text{H3K27ac(5199)}}$ on those merged peaks were used for unsupervised k-means cluster analysis.

**Pathway analysis.** The Group-1 genes identified from cluster analysis were imported into ingenuity pathway analysis (IPA) program. A list of affected pathways was calculated based on default setting.

**Reporter assay.** DNA fragments tested in a reporter assay are named as reporter fragments (R1–R10). Those fragments were inserted upstream of the SV40 promoter and Firefly luciferase in a pGL3 promoter vector. For each transfection reaction, 100 ng control plasmid expressing *Renilla* luciferase and 300 ng Firefly luciferase construct were co-transfected into $2 \times 10^5$ cells in a 24-well plate well. After 24 h, luciferase activities were measured by the Dual-Luciferase Reporter Assay System (Promega, E1910).

**Chromatin conformation capture assay.** The experiment was performed as described [52]. Briefly, cells were crosslinked with 1% formaldehyde at room temperature for 10 min, quenched with 0.125 M glycine, lysed, and treated with 600 U HindIII (NEB, #R3104) at 37 °C overnight followed by a 4 h ligation with T4 enzyme (NEB, M0202L) at 16 °C. Ligated products were quantified in triplicate by TaqMan real-time PCR. Probes and primers (listed in Supplementary Table 2) were designed by using primer blast provided by NCBI. Control 3C template was generated by using two bacterial artificial chromosomes (BACs), 656G14 and 1125P18, which together encompass putative K-M enhancer and *MGMT* promoter regions. Equimolar of the two BACs were digested with HindIII and ligated. The ligation product from BAC control was used for normalization. The relative interaction frequency was calculated as: $2^{\text{Ct(BAC)}-\text{Ct(3C)}}$

**Immunofluorescence.** Since the enhancer is localized between *MKI67* and *MGMT*, we evaluated both MGMT and Ki67 expression in patient samples using immunofluorescence. OCT-embedded patient tumor tissues were sectioned at 5-micron thickness. Slides were fixed by 4% paraformaldehyde at 4 °C for 10 min, penetrated with PBS plus 0.25% Triton X-100 for 10 min, and then treated with steam TBS antigen retrieval buffer at pH 9.0 for 60 min. One hour primary antibody incubation (1:100 dilution) was performed at room temperature followed by 1 h secondary antibody incubation (1:200 dilution). Slides were rinsed, dehydrated, and mounted with Prolong Gold antifade mounting media with DAPI (Invitrogen, Cat #P36935) and analyzed by confocal microscopy (LSM 780; ×63 objective lenses). MGMT and Ki67 were considered positive when uniform staining was detected in cell nuclei. CD45 was considered positive when cytoplasmic staining was detected.

The signal intensity for each cell was quantified in ImageJ program. The area of each nucleus was determined by DAPI staining. The total signal for each nucleus was calculated as: signal = area × average signal intensity. The total signal for 100 nuclei from each slides were quantified and plotted.

The investigator was blinded to sample allocation during immunofluorescence image collection and counting.

**Guide RNA design and cloning.** All guide RNAs were designed by using MIT CRISPR Design website (http://crispr.mit.edu). To minimize potential off-target effects of guide RNA, only high-score guide RNAs (score >85) were used. Guide RNA sequences are listed in Supplementary Table 3.

Guide RNAs used in the CRISPR/dCas9 system were cloned into lentiGuide puro vector (Addgene, Plasmid #52963)[53]. Guide RNAs used in CRISPR/Cas9 were cloned into lentiCRISPRv2 vector using the same protocol.

**Enhancer activation by CRISPR/dCas9$^{\text{p300 Core}}$ system.** A total of 5199 cells were infected with a mixture of virus containing five-guide RNAs while HEK293T cells were co-transfected with a pooled guide RNA containing five-guide RNA constructs. Both 5199 cell and HEK293T are transfected with a pcDNA-dCas9-p300 Core (or control dCas9) plasmid (Addgene, Plasmid #61357). Puromycin selection was performed 24 h post transfection. Targeting of dCas9$^{\text{p300 Core}}$ protein was confirmed by Flag ChIP-qPCR assay at 72 h post transfection. Enhancer activity and *MGMT* transcript were assessed by H3K27ac ChIP-qPCR and RT-PCR, respectively. Primers used for ChIP-qPCR and RT-PCR assay are listed in Supplementary Table 2.

**CRISPR/Cas9-mediated genomic deletion**. Guide RNAs were cloned into lentiCRISPR v2 vector (Addgene, Plasmid #52961). Lentiviruses for CRISPR editing were produced in HEK293T cells. In the genomic deletion experiment, SKMG3 and 3080 cells were infected with equal amount of lentivirus carrying two-guide RNAs flanking the region to be deleted, followed by clonal selection under puromycin and clone expansion. Paired guide RNAs g1/g2 were used to generate larger deletion, while paired guide RNAs g1/g3 and g2/g3 were used to generate the smaller deletions, respectively. PCR amplification was used for genotypic characterization of putative deletion clones. The sequence of those primers is listed in supplemental table (Supplementary Table 2). The PCR products of positive clones with homozygous deletion were validated by Sanger sequencing. The 3080 deletion clones consistently grew poorly in culture. For this reason, the initial deletion clones were implanted into the flank of nude mice, and from this effort, we obtained a single homozygous clone that subsequently grew and could be tested in vitro. This work is conducted under relevant ethical regulations of Mayo Clinic Institutional Animal Care and Use Committee (IACUC protocol number: A00003130-17).

**Clonogenic assay**. Clonogenic assays were performed to assess the effect of the deleted enhancer on TMZ sensitivity. Briefly, SKMG3 parental cells and enhancer deletion clones were plated in 6-well plates (250 cells/well), treated with graded concentration of TMZ in presence or absence of $O_6BG$ and were cultured for 2 weeks. Colonies were fixed and stained with crystal violet (0.005% (w/v) Crystal violet, 25% (v/v) Methanol). Colonies with >50 cells were manually counted, $IC_{50}$ values were calculated in GraphPad Prism 7 using a multiparametric nonlinear regression model.

**Neurosphere assay**. Primary cells suspended in StemPro NSC media were plated in triplicate in 96-well plates (500 cells per well) and treated with graded concentration of TMZ (0–1000 μM/L final concentration). Intact neurospheres containing more than 50 cells were counted after 15 days. Cell viability was calculated relative to DMSO control. $IC_{50}$ values were calculated as above.

**Cell proliferation assay**. IncuCyte Live Cell Analysis system was used to measure cell proliferation of SKMG3 wild-type clone and K-M enhancer-deleted clones. Each clone was plated into triplicate wells of a 96-well plate at 500 cells per well. The percent confluency for each well was measured every 4 h for 6 days. The confluency data were used to calculate a proliferation rate as the fold change in confluency each day as compare to Day 0.

CellTiter-Blue® Cell Viability Assay kit (CTB) was used to measure cell viability for SKMG3 and SKMG3-Del1 clones. Each clone was plated into triplicate wells of a 96-well plate at 1000 cells per well. Cell metabolism rate was measured every other day according to kit manufacturer's instruction. Fluorescence signal was read by GloMax®-Muti Microplate Multimode Reader with excitation at 560 nm and emission at 590 nm.

**Flank tumor injection**. A total of $1 \times 10^6$ cells were injected into the flank of athymic mice. Tumors were measured with calipers three times a week, and mice were killed and tumors harvested when the tumors exceed 2000 mm³.

**Statistical analysis**. ChIP-qPCR analysis for patient samples was performed only once due to limited tissue availability. For all other results, three independent experiments were performed.

A two-tailed Student's *t* test was used to establish statistical significance between control and testing group for all comparison between two data sets. One-way ANOVA was used for MGMT and Ki67 signal intensity comparison analysis.

**Data availability**. The data that support the findings of this study are available from the article and Supplementary Information Files, or from the corresponding author on request. ChIP-seq and RNA-seq data sets used this study are publically available in the GEO database, under the GSE accession number GSE113816.

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

## Acknowledgements

We thank Shulan Tian for help with statistical analysis. We thank Mark A. Schroeder and Brett L. Carlson for help with mouse work. We thank Shiv K. Gupta and Danielle M. Burgenske for helpful comments. This work was funded by National Institutes of Health (NIH) grants CA184320, CA176830, CA108961 to J.N.S. and CA157489 and CA204297 to Z.Z.

## Author contributions

X.C., J.N.S. and Z.Z. conceived and designed the experiments. X.C. performed the majority of the experiments. M.Z. made all Del1 deletion clones, tested cell growth of SKMG3-Del1 clones, and P300 targeting assay on 5199 cells. H.G. performed all of the bioinformatics analysis except pathway analysis. X.C. and H.W. performed the ChIP-qPCR analysis for Fig. 2a together. J.L. and X.C. performed the ChIP-seq assay. X.C. and D.F. cloned the reporter plasmid and guide RNA plasmid. G.K. did ingenuity pathway analysis and provided PDX tissues for experiments. L.H. and Z.H. did flank tumor injection and monitored tumor growth. I.J.P. and F.B.M. contributes critical reagents. C.G. reviewed all patient slides and was served as the director of NeuroOncology biobank.

## Additional information

**Competing interests:** The authors declare the following: a preliminary patent application was filed based on the current studies. The authors declare no other competing interests.

