## [Peer Review File · Nature Communications]

Reviewers' comments:

Reviewer #1 (Remarks to the Author):

Over all this is a very interesting, carefully prepared study providing new insight into a mechanism that may contribute to acquired resistance to TMZ therapy in MGMT methylated GBM. The mechanism of enhancer activation for re-expression of genes despite promoter methylation is of general relevance.

The mechanisms are worked out well for re-expression of MGMT and the effect on expression of Ki67. However, the role for Ki67 is less clear. It is usually "just" used as a marker for proliferation – but in the present context, its actual function during the cell cycle would deserve some attention.

Comments

- Accession number for ChIP-seq data is missing
- "Ki67 staining is associated with elevated aggressiveness of glioblastoma". Ki67 is a marker of proliferation that in GBM is not established as prognostic marker (see WHO classification 2016 and respective references). The added references are irrelevant, they are not on GBM.
- "H3K9me2/3 increases globally in GBM cells upon TMZ treatment" reference 28, Braig et al has nothing to do with this statement. This statement needs an appropriate reference/support!
- That the observed upregulated pathways may contribute to the aggressiveness of the tumors is possible, but whether they contribute directly to TMZ resistance – which is a specific effect – is another question. This should be made clear.
- Table 1 would profit from showing the associated fold change and statistics, corrected for multiple testing. Suppl Table1, add the p-value and adjusted p-value for all comparisons. Add the corresponding gene symbols of the nearest gene. Mention Table S1 in the first part of the Results together with Table 1.
- Re-expression of MGMT in the recurrent tumor of patient 1: What is the percentage of cells positive for MGMT? It looks like only 10% based Fig 3 & Fig S4. Please add the estimation for the whole section. What does this low frequency mean for the detection of an "active" enhancer? Please comment.
- Based on the effort in showing co-expression of Ki67 and MGMT I am wondering whether the authors are expecting that MGMT should only be expressed in Ki67 positive cells? Ki67 is normally only expressed during part of the cell-cycle (which makes it a proliferation marker). Are the authors proposing that the (forced) activated enhancer will change this pattern? Then cell cycle analysis should clarify this point (e.g. ImageStream) or other mechanistic evaluations.
- "Reduction of aggressiveness...", what should this mean in the context of Ki67 – there is no evaluation or even mention of the function of the protein in the cell cycle.
- There is initial and acquired TMZ resistance beyond MGMT. Furthermore, the authors show that other enhancers are affected according to their analyses.

Minor comments:

- Reference 1 (Stupp et al) is duplicated
- Introduction: "MGMT expression is regulated... » MGMT is de-regulated in cancer by methylation, in normal tissue it is not methylated, should be corrected
- "Table 1 description, it is not clear what the following means: "Genes were sorted from top to bottom based on the ratio of its expression in 3080 line divided by its expression in 5199 line". Do you mean "clone" (not line of the table)?
- Fig 3d add "...proliferating cells" in the label of the y-axes
- Brett Ref, missing
- The argumentation using citations from the literature to justify the cause/results are sometimes a little simplistic, both in the introduction and the discussion, some editing would improve the manuscript.
- Some proof reading is required

Reviewer #2 (Remarks to the Author):

The manuscript "A novel enhancer..." submitted by Chen et al describes the identification of a novel enhancer element that regulates expression of both MGMT and Ki67 and when activated can overcome MGMT promoter methylation and drive MGMT expression and TMZ resistance. There are a number of positives associated with this work. First, the work is novel and highly significant. Although MGMT promoter methylation is used as a surrogate for MGMT expression, there are roughly 25% of cases in which MGMT promoter methylation and expression do not match. The work here offers an elegant explanation of this discrepancy. Additionally, as biomarkers of TMZ sensitivity have proven to be useful in the treatment of glioma, this work offers a second potentially clinically useful biomarker of TMZ response. Second, the work presented is of very high quality, and the depth of studies presented are more than necessary to answer the questions posed and to prove the existence and importance of the KM enhancer. Finally, the findings presented have been confirmed in several cell lines as well as in patient material and there is little question about the validity of the findings.

There are only a few minor points that if addressed might add to the work:

1. There are a few instances of what appear to be typos (was on line 135, surrounding on line 163)
2. A better description or map of how the F4 and R7 fragments overlap in Fig 2 would be useful
3. The analysis of patient samples on lines 188-204 is, as noted, very limited and a more complete answer would likely require some sort of single cell analysis. This would be great but is to my eye not necessary given the thoroughness of the rest of the study.
4. One could conceivably also as a second negative control use dCas9 to modify the KM enhancer in ways not expected to alter enhancer activity but again this seems to be beyond what is required
5. It was also unclear why enhancer deletion studies were done primarily in SKMG3 cells rather than the 5199 and 3080 cells in which the majority of the work was performed

Reviewer #3 (Remarks to the Author):

Chen et al report the discovery of an enhancer element that controls the expression of MGMT that plays important roles in conferring resistance to the first line therapeutic agent TMZ in glioblastoma. The authors created maps of enhancers in TMZ-resistant and TMZ-sensitive glioblastoma cells using ChIP-Seq, and identified a region occupied by the enhancer-associated histone mark H3K27Ac specifically in TMZ-resistant cells upstream of the MGMT gene. H3K27Ac enrichment at this site correlates with MGMT expression and TMZ resistance in clinical specimens. Deletion of the enhancer in TMZ-resistant cells compromises MGMT expression and renders cells sensitive to TMZ. Activation of the enhancer in TMZ-sensitive cells induces MGMT expression and confers TMZ resistance.

The study reports an important mechanism of acquired drug resistance in tumor cells, and suggests new therapeutic approaches to combat resistance through transcriptional inhibition. The experiments appear thoroughly designed and executed. I have a few suggestions to further improve the clarity and interpretation of some of the findings.

Minor comments

1) The authors make a compelling case that the region they identified is indeed an enhancer element that controls MGMT expression. Their interpretation of some of the data at a few instances is slightly inaccurate, which is distracting to the readers. Below I list these instances and offer some improvements:

1A) The authors create ChIP-Seq maps of various histone modifications in TMZ-sensitive and TMZ-resistant cells, and then claim to observe global reduction in the signal intensity of some of these marks.

(E.g. "Interestingly the levels of H3K4me1 and H3K27ac at the enhancer regions were globally reduced in the 3080 line compared to the 5199 line. Moreover, the level of H3K9me3 was increased at the transcription starting sites (TSS) in the TMZ-resistant 3080 compared to the TMZ-sensitive 5199 (Fig. S1)."

This type of analysis is only possible when performing ChIP-Seq using internal controls such as precise amounts of spiked-in fragmented chromatin of other species (ChIP-RX, Orlando et al, Cell Reports 2015). Without the spiked-in internal control, it is not possible to rule out that any change in global levels is not caused by different quality of ChIP in the different samples. This type of global analysis is not necessary for the authors, and I do not suggest to repeat the ChIP experiments using internal controls. Instead, the authors are advised to report the numbers of peaks they identify in the data, and the correlation of the read densities in those regions in the TMZ-sensitive and TMZ-resistance cells. This shall give a reliable and comprehensive view of the differences in the enhancer landscape in these cells

1B) The authors perform luciferase reporter assays to demonstrate that the enhancer element they identify has enhancer reporter activity. The assays reveal that fragments of the region they identify display enhancer reporter activity in these assays, but not all of them. The authors interpret this as the following:

"These results suggest that this region can enhance gene transcription and the R7 region may act as an MGMT enhancer, whereas other regions may serve as enhancer for other genes such as Ki67." (line 150-152)

The latter interpretations is likely incorrect. What the data suggests is that the R7 region (and R1 and R10 based on the data in Figure 2C) display enhancer activity in the reporter assay, and I recommend that authors state this. The authors used the SV40 promoter in the reporter system, which is by its nature artificial to begin with, so the data does not inform on which gene promoters the enhancer activates in vivo. To do that, the authors would have needed to clone the MGMT (or Ki67) promoter in the reporter vector. In my view, rephrasing this sentence as noted above is sufficient.

2) The data implicates the enhancer element discovered by the authors in the control of the MGMT and Ki67 genes, but whether the enhancer controls MGMT only or both MGMT and Ki67 is unclear. There is much emerging evidence that chromosome structure plays important roles in constraining enhancers to operate on specific genes. The genome is organized into CTCF-CTCF loops, that contribute to larger Topologically Associating Domains (TADs). Enhancers that occur within such CTCF-CTCF loops or TADs predominantly activate genes within the same loop/TAD. It would be very valuable to observe the 3D organization of the Ki67-enhancer-MGMT locus, which would inform on the likely physiological targets of the enhancer. Getting chromatin contact data in these cells would be quite complex, and in my view not necessary for this study, but the authors can explore published reference Hi-C (Rao et al, Cell 2014) or ChIA-PET datasets (Ji et al, Cell Stem Cell 2016, Tang et al, Cell 2015) for this purpose.

3) The authors propose that combination therapy with TMZ and transcription inhibitors may suppress the emergence of TMZ-resistance in glioblastoma. In addition to the BET inhibitors they mention, the authors may want to reference and suggest using CDK7 inhibitors to suppress emergence of drug resistance, as demonstrated in a recent study (Rusan et al, Cancer Discovery 2018).

4) Typos:

Line 117: Moreover

Line 185: PDX

Line 325: Idea

Response to referees:

Reviewers' comments:

Reviewer #1 (Remarks to the Author):

Over all this is a very interesting, carefully prepared study providing new insight into a mechanism that may contribute to acquired resistance to TMZ therapy in MGMT methylated GBM. The mechanism of enhancer activation for re-expression of genes despite promoter methylation is of general relevance. The mechanisms are worked out well for re-expression of MGMT and the effect on expression of Ki67. However, the role for Ki67 is less clear. It is usually “just” used as a marker for proliferation – but in the present context, its actual function during the cell cycle would deserve some attention.

Response: We thank the reviewer for their very positive comments about this exciting story. We have addressed each concern of the reviewer as detailed below.

Reviewer #1

Comments

- *Accession number for ChIP-seq data is missing*

Response: We have submitted the ChIP-seq and RNA-seq datasets to GEO and the accession number for these datasets is now listed at the end of the methods section on page 31.

Reviewer #1

- *“Ki67 staining is associated with elevated aggressiveness of glioblastoma”. Ki67 is a marker of proliferation that in GBM is not established as prognostic marker (see WHO classification 2016 and respective references). The added references are irrelevant, they are not on GBM.*

Response: We agree with the reviewer that the references cited previously as well as the statement were not directly relevant to GBM. We have edited the statement and added additional references specific to GBM. We also modified the text on page 4 as below:

“In other cancers such as breast cancer¹⁻³, the fraction of cells staining positive for Ki67 is associated with increased proliferation and an adverse clinical outcome. While more controversial in GBM with typically intermediate levels of Ki67 expression⁴, high Ki67 staining is associated with elevated proliferation and poor prognosis of brain tumors in some of these studies⁴⁻⁷.”

Reviewer #1

- *“H3K9me2/3 increases globally in GBM cells upon TMZ treatment” reference 28, Braig et al has nothing to do with this statement. This statement needs an appropriate reference/support!*

Response: We are sorry for our oversight. We modified the statement as the following and cited the correct reference⁸ on page 6.

The new reference is: Papait, R., Magrassi, L., Rigamonti, D. & Cattaneo, E. Temozolomide and carmustine cause large-scale heterochromatin reorganization in glioma cells. *Biochemical and Biophysical Research Communications* 379, 434-439, doi:<https://doi.org/10.1016/j.bbrc.2008.12.09>

The text on page 6 was modified as: “We note a previous report showing that heterochromatin reorganizes and H3K9me2/3 increases in GBM cells treated with TMZ²⁸.”

Reviewer #1

• *That the observed upregulated pathways may contribute to the aggressiveness of the tumors is possible, but whether they contribute directly to TMZ resistance – which is a specific effect – is another question. This should be made clear.*

Response: To address this concern, we revised the text on page 6 as follow:

“Pathway analysis of this group of genes indicated that genes involved in gliomagenesis and cancer drug resistance are enriched in this group (Fig. 1i). Interestingly, MGMT, a key driver of TMZ resistance⁹, was one of the top 10 genes in the Group-1 gene list (Table 1 and Table S2). These results suggest that a subgroup of enhancers was activated in the 3080 line, and at least one of these directly contribute to TMZ resistance”.

Reviewer #1

• *Table 1 would profit from showing the associated fold change and statistics, corrected for multiple testing. Suppl Table1, add the p-value and adjusted p-value for all comparisons. Add the corresponding gene symbols of the nearest gene. Mention Table S1 in the first part of the Results together with Table 1.*

Response: As the reviewer suggested, we made 3 major changes to our figures and tables.

First, we added associated fold change, p-value and adjusted p-value for gene expression analysis by RNA-seq in Table 1 and Table S1 (Table S2 in the revised manuscript). Second, we added the p-value and gene symbols of the nearest gene in Table S1. Since the programs used for analysis of CHIP-seq datasets do not give adjusted p value, we did not include adjusted p value in this Table. Third, we modified the text on page 7 to discuss the data in table S1 together with data from Table 1 as follow:

“Interestingly, MGMT, a key driver of TMZ resistance²⁹, was one of the top 10 genes in this list that shows significant changes in gene expression (Table 1 and Supplementary Table 2).”

Reviewer #1

• *Re-expression of MGMT in the recurrent tumor of patient 1: What is the percentage of cells positive for MGMT? It looks like only 10% based Fig 3 & Fig S4. Please add the estimation*

for the whole section. What does this low frequency mean for the detection of an “active” enhancer? Please comment.

Response: To answer this question, we calculated the percentage of MGMT positive cells based on MGMT and DAPI staining in patient samples as well as in two PDX lines, 5199 and 3080. As shown in Letter Figure 1, the percentage of cells positive for MGMT in the patient 1 recurrent sample is 24%, which is close to that of the TMZ-resistant 3080 PDX line studied in our experimental conditions.

Letter Figure 1. Analysis of MGMT expression in patient samples. One hundred cells are chosen randomly from each slide for analysis. MGMT expression level in each nucleus was analyzed based on the MGMT intensity (Intensity= average intensity \times nucleus size, nucleus size is determined by DAPI staining). The cutoff line for MGMT positive staining was set based on the assumption that most cells in the 5199 line do not express MGMT.

The relatively low percentage of MGMT positive cells detected in tumor samples and the positive control sample is likely due to tumor heterogeneity. Glioblastoma is known for its extensive intratumor heterogeneity¹⁰, which appears to extend to MGMT expression. Supporting this idea, in an analysis of 50 GBM patient samples, only a single sample had greater than 50% MGMT positive cells and most MGMT expressing tumor samples had only 10-50% MGMT positive cells¹¹.

To better present our results, we have added a sentence on page 10 as listed below to describe the percentage of MGMT positive cells in the patient recurrent samples and benchmarked against our experimental models 3080 and 5199 sub-lines. We have added this as Supplementary figure S4.

The text was modified as:

“Based on the immunofluorescence staining, we estimated that the fraction of MGMT expressing cells increased from 4% in the primary tumor of patient #1 to 24% in recurrent

tumor of this patient. A similar percentage of MGMT expressing cells were also detected in the 3080 subline (data not shown). The relatively low percentage of MGMT positive cells detected in patient #1 and in 3080 subline is likely due to tumor heterogeneity and/or relatively low sensitivity of MGMT immunofluorescence. Supporting this idea, glioblastoma is known for its extensive intratumor heterogeneity³³. It has been shown previously that only 1 out of 50 patient samples had over 50% MGMT positive cells, and most MGMT expressed tumor samples have 10-50% MGMT positive cells³⁴. The fraction of MGMT expressing proliferating cells was not increased in recurrent tumors of patient #2 and #3 compared to their corresponding primary tumors (Fig. 3d-f, Supplementary Fig. 5).”

Reviewer #1

• *Based on the effort in showing co-expression of Ki67 and MGMT I am wondering whether the authors are expecting that MGMT should only be expressed in Ki67 positive cells?*

Response: The rationale to co-stain for MGMT and Ki67 was based on the observation that both genes are located near the enhancer. As described below, Ki67 and MGMT co-localize in the same TADs in neuroblastoma cells, but Ki67 is localized at boundary of the TAD containing MGMT in normal brain cells (Fig. 6b-c), suggesting that co-expression of Ki67 and MGMT may be cell type specific. In fact, deletion of the K-M enhancer reduces the MGMT expression more dramatically than the expression of Ki67 (Figure 6e). In the revised manuscript, we made these points clear when we describe Fig. 6 on page 14. The related description in revised manuscript is:

“*MKI67* is a gene encoding the nuclear protein Ki67, which is a proliferation marker for many tumors including GBM (Supplementary Fig. 8). Because *MKI67* is another gene localized close to the enhancer, we first analyzed the published Hi-C datasets³⁸ to determine whether the *MKI67* gene localized in the same topologically associating domain (TAD) with MGMT. We found that *MKI67* resided in the same TAD as MGMT and the enhancer in neuroblastoma cells (Fig. 6b) and localized at the left boundary of the TAD domain containing MGMT in cortical and subcortical plate cells (Fig. 6c), suggesting that the K-M enhancer may also regulate Ki67 in a cell type dependent manner. We then analyzed Ki67 expression in the enhancer deletion clones. We observed that two SKMG3 clones with the 3.3 kb deletion, which show reduced proliferation (Fig. 6d), exhibited a significant decrease in Ki67 expression (Fig. 6e). Tumor growth rate and Ki67 expression were also reduced in the 3080 Del1 deleted clone (Fig. 6f-g). However, we noticed that in all these clones with the deletion of the enhancer element, the expression of MGMT was affected more than Ki67. Collectively, our results indicate that the K-M enhancer is the enhancer for MGMT that likely also regulates the expression of Ki67 in some cell type, which in turn regulates TMZ sensitivity and possibly cell proliferation.”

Reviewer #1

Ki67 is normally only expressed during part of the cell-cycle (which makes it a proliferation marker). Are the authors proposing that the (forced) activated enhancer will change this

pattern? Then cell cycle analysis should clarify this point (e.g. ImageStream) or other mechanistic evaluations.

Response:

While Ki67 is a known proliferation mark, the function of Ki67 is less well understood. A recent study indicates that Ki67 can serve as a surfactant to separate mitotic chromosomes¹². In other reports, knockdown of Ki67 in cell lines results in mild growth defect, while cell cycle is not affected^{13,14}. Similarly, knockout of Ki67 in mice has no apparent effect on cell growth and survival¹⁵. In SKMG3 lines, deletion of the MGMT enhancer only led to a mild (30%-50%) reduction of Ki67 expression (Fig. 6e). Therefore, deletion of the enhancer should not affect cell cycle. Indeed, our flow cytometry analysis of DNA content shows that deletion of the K-M enhancer does not affect cell cycle phases (Letter Figure 2). In the revised manuscript, we discuss these points in the discussion on page 13-14 as:

“However, cell cycle appears to be normal in the enhancer deleted clones (data not shown).”

Letter Figure 2. K-M enhancer deletion does not alter cell cycle. Flow cytometry analysis of DNA content was performed on SKMG3 parental line, SKMG3 wild type clone and K-M enhancer deleted SKMG3 clones. Percentage of cells at each phase of the cell cycle is calculated and error bar indicates standard deviation obtained from 3 biological repeats.

Reviewer #1

• “Reduction of aggressiveness...”, what should this mean in the context of Ki67 – there is no evaluation or even mention of the function of the protein in the cell cycle.

Response: We agree that more Ki67 background should be given in the introduction section. To address this question, we modified the text on page 4 with additional references.

Reviewer #1

• There is initial and acquired TMZ resistance beyond MGMT. Furthermore, the authors show that other enhancers are affected according to their analyses.

Response: We absolutely agree that there are multiple mechanisms related to intrinsic and acquired resistance to TMZ beyond the effects on MGMT. With multiple enhancers altered in TMZ resistant GBM12 3080 model, we are very interested in investigating whether other enhancers drive alternative mechanisms that contribute to TMZ resistance in this line or in other lines. We will test these ideas in the future. This point is now discussed in the Discussion section on page 15-16 as:

“Therefore, we speculate that in addition to genetic mutations, enhancer alterations through changes in chromatin states likely contribute both to intrinsic and acquired TMZ resistance.”

Minor comments:

- Reference 1 (Stupp et al) is duplicated

Response: This duplicated reference has been removed from current manuscript.

Reviewer #1

- *Introduction: “MGMT expression is regulated.... » MGMT is de-regulated in cancer by methylation, in normal tissue it is not methylated, should be corrected*

Response: We have corrected the text on page 3 as “MGMT expression can be silenced by the methylation of a promoter/enhancer (P/E) region, which contains a promoter and a 59 bp cis-acting enhancer element that spans the first exon-intron boundary of MGMT gene”.

Reviewer #1

- *“Table 1 description, it is not clear what the following means: “Genes were sorted from top to bottom based on the ratio of its expression in 3080 line divided by its expression in 5199 line”. Do you mean “clone” (not line of the table)?*

Response: To make our statement more clear, we have clarified the table legend of table 1 as below:

Table 1. The top 10 nearby genes with the most altered Group-1 enhancers. Genes with the most elevated expression within the 1141 genes in Group-1 are listed in the table. Associated fold changes represents the fold change in gene expression in the 3080 subline over 5199 line (as calculated by normalized RNA-seq reads in 3080 line divided by normalized RNA-seq reads in 5199 line). Genes are sorted by fold change and 10 genes with the highest fold change are presented in the table. The p value and adjusted p value are also shown.

Reviewer #1

- Fig 3d add “...proliferating cells” in the label of the y-axes

Response: This panel has been modified in the new manuscript.

Reviewer #1

- Brett Ref, missing

Response: This reference has been added on method section at page 19.

Reviewer #1

• *The argumentation using citations from the literature to justify the cause/results are sometimes a little simplistic, both in the introduction and the discussion, some editing would improve the manuscript.*

Response: To establish a better cause/ results relationship for our introduction and discussion, we have modified the introduction and discussion with more details and added more references.

Reviewer #1

- Some proof reading is required

Response: We have now carefully proofread the manuscript and believe we have identified and fixed all of the typographic and grammatical errors.

Reviewer #2 (Remarks to the Author):

The manuscript “A novel enhancer...” submitted by Chen et al describes the identification of a novel enhancer element that regulates expression of both MGMT and Ki67 and when activated can overcome MGMT promoter methylation and drive MGMT expression and TMZ resistance. There are a number of positives associated with this work. First, the work is novel and highly significant. Although MGMT promoter methylation is used as a surrogate for MGMT expression, there are roughly 25% of cases in which MGMT promoter methylation and expression do not match. The work here offers an elegant explanation of this discrepancy. Additionally, as biomarkers of TMZ sensitivity have proven to be useful in the treatment of glioma, this work offers a second potentially clinically useful biomarker of TMZ response. Second, the work presented is of very high quality, and the depth of studies presented are more than necessary to answer the questions posed and to prove the existence and importance of the KM enhancer. Finally, the findings presented have been confirmed in several cell lines as well as in patient material and there is little question about the validity of the findings. There are only a few minor points that if addressed might add to the work:

Response: We thank the reviewer’s very positive comments about our manuscript. We have addressed all reviewer’s comments as described below.

Reviewer #2

1. There are a few instances of what appear to be typos (was on line 135, surrounding on line 163)

Response: We are sorry for our oversight. We edited the manuscript carefully and corrected these typos in the revised manuscript.

Reviewer #2

2. A better description or map of how the F4 and R7 fragments overlap in Fig 2 would be useful

Response: To address this concern, we modified Figure 2c, d using two different color codes to show the overlap.

Reviewer #2

3. The analysis of patient samples on lines 188-204 is, as noted, very limited and a more complete answer would likely require some sort of single cell analysis.

Response: We agree with the reviewer that it is a great idea to analyze patient samples using single cell analytic techniques. In fact, this is what we proposed to do in the future in our grant proposal submitted to NIH. To address this concern, we added the following sentences. “In the future, it would be interesting to analyze the enhancer activity in a larger cohort of patient samples at single cell levels” on page 11.

Reviewer #2

4. One could conceivably also as a second negative control use dCas9 to modify the KM enhancer in ways not expected to alter enhancer activity but again this seems to be beyond what is required

Response: In the experiment, we used dCas9+sgRNAs as controls for dCas9-p300+sgRNAs. In principle, it is a good idea to target a chromatin regulator, which has no effect on the enhancer activity, using dCas9 to the enhancer region. However, it would be challenging to predict how the chromatin regulator affects locally chromatin structure. Therefore, I agree with the reviewer that this kind of control, while interesting, is beyond the scope of the present manuscript.

Reviewer #2

5. It was also unclear why enhancer deletion studies were done primarily in SKMG3 cells rather than the 5199 and 3080 cells in which the majority of the work was performed

Response: We created enhancer deletion clones in both SKMG3 and 3080 cells. In SKMG3 line, we obtained three independent clones with homozygous deletion of the enhancer, and while growth was heterogeneously affected in these clones, they all proliferated in cell culture at a level that supported robust experimental evaluation across multiple SKMG3 clones to account for possible clonal variation (Figure 5i, j and Figure 6f, g). In contrast, the 3080 deletion clones consistently grew poorly in culture. For this reason, the initial deletion clones were implanted into the flank of nude mice, and from this effort, we obtained a single

homozygous clone that subsequently grew and could be tested in vitro. Using this clone, we validated the SKMG3 results as detailed in Figure 6. In the revised manuscript, we made this clear in the Methods Section on page 26.

Reviewer #3(Remarks to the Author):

Chen et al report the discovery of an enhancer element that controls the expression of MGMT that plays important roles in conferring resistance to the first line therapeutic agent TMZ in glioblastoma. The authors created maps of enhancers in TMZ-resistant and TMZ-sensitive glioblastoma cells using ChIP-Seq, and identified a region occupied by the enhancer-associated histone mark H3K27Ac specifically in TMZ-resistant cells upstream of the MGMT gene. H3K27Ac enrichment at this site correlates with MGMT expression and TMZ resistance in clinical specimens. Deletion of the enhancer in TMZ-resistant cells compromises MGMT expression and renders cells sensitive to TMZ. Activation of the enhancer in TMZ-sensitive cells induces MGMT expression and confers TMZ resistance. The study reports an important mechanism of acquired drug resistance in tumor cells, and suggests new therapeutic approaches to combat resistance through transcriptional inhibition. The experiments appear thoroughly designed and executed. I have a few suggestions to further improve the clarity and interpretation of some of the findings.

Response: We thank the reviewer's very positive comments about our manuscript. We have addressed all of the reviewer's minor concerns as detailed below.

Reviewer #3

Minor comments

1) The authors make a compelling case that the region they identified is indeed an enhancer element that controls MGMT expression. Their interpretation of some of the data at a few instances is slightly inaccurate, which is distracting to the readers. Below I list these instances and offer some

1A) The authors create ChIP-Seq maps of various histone modifications in TMZ-sensitive and TMZ-resistant cells, and then claim to observe global reduction in the signal intensity of some of these marks.

(E.g. "Interestingly the levels of H3K4me1 and H3K27ac at the enhancer regions were globally reduced in the 3080 line compared to the 5199 line. Moreover, the level of H3K9me3 was increased at the transcription starting sites (TSS) in the TMZ-resistant 3080 compared to the TMZ-sensitive 5199 (Fig. S1)."

This type of analysis is only possible when performing ChIP-Seq using internal controls such as precise amounts of spiked-in fragmented chromatin of other species (ChIP-RX, Orlando et al, Cell Reports 2015). Without the spiked-in internal control, it is not possible to rule out

that any change in global levels is not caused by different quality of ChIP in the different samples. This type of global analysis is not necessary for the authors, and I do not suggest to repeat the ChIP experiments using internal controls. Instead, the authors are advised to report the numbers of peaks they identify in the data, and the correlation of the read densities in those regions in the TMZ-sensitive and TMZ-resistance cells. This shall give a reliable and comprehensive view of the differences in the enhancer landscape in these cells

Response: We agree with the reviewer that it is necessary to use spike-in chromatin to compare two samples when there are large changes in histone modifications, which we have done previously for other studies (Fang et al Science 2016). However, H3K4me1 and H3K27ac not only present on enhancer regions but also present in other regions such as TSS regions, 3' UTR regions and 5' UTR regions. Although we observed that H3K4me1 and H3K27ac are reduced at enhancer regions, they are not globally reduced based on Western blot analysis (new Figure 1b). Therefore, we believe some comment on global changes can be made without use of spike-in chromatin. In revised manuscript, we carefully point out these ideas on line 105. We also performed the analysis suggested by the reviewer. The number of peaks we identified is listed in Table S1. The correlation of the read densities are shown on Letter Figure 3.

Letter Figure 3. The correlation of ChIP-seq peaks in 5199 and 3080 lines. ChIP-seq peaks were identified for each histone mark. Normalized Log2 ChIP-seq reads densities of each peak in the 5199 and 3080 line were plotted on X- and Y-axis, respectively.

Reviewer #3

1B) The authors perform luciferase reporter assays to demonstrate that the enhancer element they identify has enhancer reporter activity. The assays reveal that fragments of the region they identify display enhancer reporter activity in these assays, but not all of them. The authors interpret this as the following:

“These results suggest that this region can enhance gene transcription and the R7 region may act as an MGMT enhancer, whereas other regions may serve as enhancer for other genes such as Ki67.” (line 150-152)

The latter interpretation is likely incorrect. What the data suggests is that the R7 region (and R1 and R10 based on the data in Figure 2C) display enhancer activity in the reporter assay, and I recommend that authors state this. The authors used the SV40 promoter in the reporter system, which is by its nature artificial to begin with, so the data does not inform on which gene promoters the enhancer activates in vivo. To do that, the authors would have needed to clone the MGMT (or Ki67) promoter in the reporter vector. In my view, rephrasing this sentence as noted above is sufficient.

Response: We agree with the reviewer that our reporter assay data is not sufficient to definitively show R7 region can serve as an enhancer for Ki67. We modified the text on page 8 as “These results suggest that R1, R7 and R10 regions can enhance gene transcription from the SV40 promoter”.

Reviewer #3

2) The data implicates the enhancer element discovered by the authors in the control of the MGMT and Ki67 genes, but whether the enhancer controls MGMT only or both MGMT and Ki67 is unclear. There is much emerging evidence that chromosome structure plays important roles in constraining enhancers to operate on specific genes. The genome is organized into CTCF-CTCF loops, that contribute to larger Topologically Associating Domains (TADs). Enhancers that occur within such CTCF-CTCF loops or TADs predominantly activate genes within the same loop/TAD. It would be very valuable to observe the 3D organization of the Ki67-enhancer-MGMT locus, which would inform on the likely physiological targets of the enhancer. Getting chromatin contact data in these cells would be quite complex, and in my view not necessary for this study, but the authors can explore published reference Hi-C (Rao et al, Cell 2014) or ChIA-PET datasets (Ji et al, Cell Stem Cell 2016, Tang et al, Cell 2015) for this purpose.

Response: We followed the reviewer’s suggestion and examined several Hi-C datasets. We found that in neuroblastoma cells, Ki67, K-M enhancer and MGMT are located at the same TAD (Fig. 6b). In cortical plate, Ki67 is located at the right boundary of TADs that contain MGMT and K-M enhancer (Fig. 6c). These results suggest that the enhancer can regulate the expression of MGMT in most cell lines, and may also regulate Ki67 in a cell type dependent

manner. This finding is consistent with the differential effect of deletion of the K-M enhancer on the expression of MGMT and Ki67, suggesting that this enhancer primarily regulates MGMT expression. We included this result in Fig. 6 and make these points clear in the revised manuscript on page 14 as follow:

“*MKI67* is a gene encoding the nuclear protein Ki67, which is a proliferation marker for many tumors including GBM (Supplementary Fig. 8). Because *MKI67* is another gene localized close to the enhancer, we first analyzed the published Hi-C datasets³⁸ to determine whether the *MKI67* gene localized in the same topologically associating domain (TAD) with MGMT. We found that *MKI67* resided in the same TAD as MGMT and the enhancer in neuroblastoma cells (Fig. 6b) and localized at the left boundary of the TAD domain containing MGMT in cortical and subcortical plate cells (Fig. 6c), suggesting that the K-M enhancer may also regulate Ki67 in a cell type dependent manner. We then analyzed Ki67 expression in the enhancer deletion clones. We observed that two SKMG3 clones with the 3.3 kb deletion, which show reduced proliferation (Fig. 6d), exhibited a significant decrease in Ki67 expression (Fig. 6e). Tumor growth rate and Ki67 expression were also reduced in the 3080 Del1 deleted clone (Fig. 6f-g). However, we noticed that in all these clones with the deletion of the enhancer element, the expression of MGMT was affected more than Ki67. Collectively, our results indicate that the K-M enhancer is the enhancer for MGMT that likely also regulates the expression of Ki67 in some cell type, which in turn regulates TMZ sensitivity and possibly cell proliferation.”

3) The authors propose that combination therapy with TMZ and transcription inhibitors may suppress the emergence of TMZ-resistance in glioblastoma. In addition to the BET inhibitors they mention, the authors may want to reference and suggest using CDK7 inhibitors to suppress emergence of drug resistance, as demonstrated in a recent study (Rusan et al, Cancer Discovery 2018).

Response: We have expanded the discussion to include this interesting study on page 16-17.

4) Typos:

Line 117: Moreover

Line 185: PDX

Line 325: Idea

Response: Those typos are corrected in the current manuscript.

Reference

- 1 Inwald, E. C. *et al.* Ki-67 is a prognostic parameter in breast cancer patients: results of a large population-based cohort of a cancer registry. *Breast Cancer Research and Treatment* **139**, 539-552, doi:10.1007/s10549-013-2560-8 (2013).
- 2 Yerushalmi, R., Woods, R., Ravdin, P. M., Hayes, M. M. & Gelmon, K. A. Ki67 in breast cancer: prognostic and predictive potential. *The Lancet Oncology* **11**, 174-183, doi:[https://doi.org/10.1016/S1470-2045\(09\)70262-1](https://doi.org/10.1016/S1470-2045(09)70262-1) (2010).
- 3 de Azambuja, E. *et al.* Ki-67 as prognostic marker in early breast cancer: a meta-analysis of published studies involving 12 155 patients. *British journal of cancer* **96**, 1504, doi:10.1038/sj.bjc.6603756 (2007).
- 4 Mastronardi, L., Guiducci, A., Puzzilli, F. & Ruggeri, A. Relationship between Ki-67 labeling index and survival in high-grade glioma patients treated after surgery with tamoxifen. *Journal of neurosurgical sciences* **43**, 263-270 (1999).
- 5 Torp, S. H. Diagnostic and prognostic role of Ki67 immunostaining in human astrocytomas using four different antibodies. *Clinical neuropathology* **21**, 252-257 (2002).
- 6 Skjulsvik, A. J., Mork, J. N., Torp, M. O. & Torp, S. H. Ki-67/MIB-1 immunostaining in a cohort of human gliomas. *International journal of clinical and experimental pathology* **7**, 8905-8910 (2014).
- 7 Chen, W. J., He, D. S., Tang, R. X., Ren, F. H. & Chen, G. Ki-67 is a valuable prognostic factor in gliomas: evidence from a systematic review and meta-analysis. *Asian Pacific journal of cancer prevention : APJCP* **16**, 411-420 (2015).
- 8 Papait, R., Magrassi, L., Rigamonti, D. & Cattaneo, E. Temozolomide and carmustine cause large-scale heterochromatin reorganization in glioma cells. *Biochemical and Biophysical Research Communications* **379**, 434-439, doi:<https://doi.org/10.1016/j.bbrc.2008.12.091> (2009).
- 9 Gerson, S. L. MGMT: its role in cancer aetiology and cancer therapeutics. *Nat Rev Cancer* **4**, 296-307, doi:10.1038/nrc1319 (2004).
- 10 Sottoriva, A. *et al.* Intratumor heterogeneity in human glioblastoma reflects cancer evolutionary dynamics. *Proceedings of the National Academy of Sciences* **110**, 4009-4014, doi:10.1073/pnas.1219747110 (2013).
- 11 Rodriguez, F. J. *et al.* MGMT immunohistochemical expression and promoter methylation in human glioblastoma. *Applied immunohistochemistry & molecular morphology : AIMM* **16**, 59-65, doi:10.1097/PAI.0b013e31802fac2f (2008).
- 12 Booth, D. G. *et al.* Ki-67 is a PP1-interacting protein that organises the mitotic chromosome periphery. *eLife* **3**, e01641, doi:10.7554/eLife.01641 (2014).
- 13 Bai, Y. *et al.* Ki-67 is overexpressed in human laryngeal carcinoma and contributes to the proliferation of HEP2 cells. *Oncology letters* **12**, 2641-2647, doi:10.3892/ol.2016.4980 (2016).
- 14 Yuan, P. *et al.* Ki-67 expression in luminal type breast cancer and its association with the clinicopathology of the cancer. *Oncology letters* **11**, 2101-2105, doi:10.3892/ol.2016.4199 (2016).
- 15 Cidado, J. *et al.* Ki-67 is required for maintenance of cancer stem cells but not cell proliferation. *Oncotarget* **7**, 6281-6293, doi:10.18632/oncotarget.7057 (2016).

REVIEWERS' COMMENTS:

Reviewer #1 (Remarks to the Author):

the authors have satisfactorily replied to my questions. The manuscript is improved and is now ready for publication.

Reviewer #2 (Remarks to the Author):

Although the authors have addressed the previous concerns, there are two areas that remain or have been created in the revision.

1. line 52- the text suggests that the association between MGMT promoter methylation and favorable TMZ outcome was the basis for the development of MGMT inhibitors. This is not accurate as O6BG was developed long before the MGMT gene and promoter were even cloned. This inaccuracy should be corrected.

2. lines 300 and 373 - text added seems to imply that Ki67 expression is a driver of proliferation when existing data suggests that it is merely a marker of proliferation. If the authors wish to contend that Ki67 drives proliferation, additional studies are required. Otherwise the text needs to be modified to eliminate the suggestion of causality.

Reviewer #3(Remarks to the Author):

The authors seemed to have missed the point on one key comment. I also have suggestions to improve the revisions the authors made based on the other comments. Below are the responses pasted in their rebuttal letter.

Reviewer #3(Remarks to the Author):

Chen et al report the discovery of an enhancer element that controls the expression of MGMT that plays important roles in conferring resistance to the first line therapeutic agent TMZ in glioblastoma. The authors created maps of enhancers in TMZ-resistant and TMZ-sensitive glioblastoma cells using ChIP-Seq, and identified a region occupied by the enhancer-associated histone mark H3K27Ac specifically in TMZ-resistant cells upstream of the MGMT gene. H3K27Ac enrichment at this site correlates with MGMT expression and TMZ resistance in clinical specimens. Deletion of the enhancer in TMZ-resistant cells compromises MGMT expression and renders cells sensitive to TMZ. Activation of the enhancer in TMZ-sensitive cells induces MGMT expression and confers TMZ resistance. The study reports an important mechanism of acquired drug resistance in tumor cells, and suggests new therapeutic approaches to combat resistance through transcriptional inhibition. The experiments appear thoroughly designed and executed. I have a few suggestions to further improve the clarity and interpretation of some of the findings.

Response: We thank the reviewer's very positive comments about our manuscript. We have addressed all of the reviewer's minor concerns as detailed below.

Reviewer #3

Minor comments

1) The authors make a compelling case that the region they identified is indeed an enhancer

element that controls MGMT expression. Their interpretation of some of the data at a few instances is slightly inaccurate, which is distracting to the readers. Below I list these instances and offer some

1A) The authors create ChIP-Seq maps of various histone modifications in TMZ-sensitive and TMZ-resistant cells, and then claim to observe global reduction in the signal intensity of some of these marks.

(E.g. "Interestingly the levels of H3K4me1 and H3K27ac at the enhancer regions were globally reduced in the 3080 line compared to the 5199 line. Moreover, the level of H3K9me3 was increased at the transcription starting sites (TSS) in the TMZ-resistant 3080 compared to the TMZ-sensitive 5199 (Fig. S1)."

This type of analysis is only possible when performing ChIP-Seq using internal controls such as precise amounts of spiked-in fragmented chromatin of other species (ChIP-RX, Orlando et al, Cell Reports 2015). Without the spiked-in internal control, it is not possible to rule out that any change in global levels is not caused by different quality of ChIP in the different samples. This type of global analysis is not necessary for the authors, and I do not suggest to repeat the ChIP experiments using internal controls. Instead, the authors are advised to report the numbers of peaks they identify in the data, and the correlation of the read densities in those regions in the TMZ-sensitive and TMZ-resistance cells. This shall give a reliable and comprehensive view of the differences in the enhancer landscape in these cells

Response: We agree with the reviewer that it is necessary to use spike-in chromatin to compare two samples when there are large changes in histone modifications, which we have done previously for other studies (Fang et al Science 2016). However, H3K4me1 and H3K27ac not only present on enhancer regions but also present in other regions such as TSS regions, 3' UTR regions and 5' URT regions. Although we observed that H3K4me1 and H3K27ac are reduced at enhancer regions, they are not globally reduced based on Western blot analysis (new Figure 1b). Therefore, we believe some comment on global changes can be made without use of spike-in chromatin. In revised manuscript, we carefully point out these ideas on line 105. We also performed the analysis suggested by the reviewer. The number of peaks we identified is listed in Table S1. The correlation of the read densities are shown on Letter Figure 3.

Reviewer Response

The authors seem to have missed the point here. Spike-in controls in ChIP are essential to determine global changes in occupancy levels, and are especially important when changes are small. The new data and the data in the references the authors now cite in fact argue that the so called global changes the authors claim to observe in this manuscript are erroneous. The authors cite their previous work (Fang et al, Science 2016) as the standard for detecting global changes in histone marks. In that paper, the authors used spike-in normalization for ChIP, and the Western blots they performed were consistent with the global changes (i.e. a reduction of signal was observed on the Western blot, AND a reduction of signal was observed in ChIP when accounting for the spike-in controls). In this paper, the authors do not use spike-ins, AND the Western blot (new Fig 1b) in fact contradicts their idea of global changes, (i.e. the levels of histone marks are unaffected).

At this point, I suggest that the authors forgo any speculation about global changes, as the experimental evidence for it is insufficient, and whether the changes are global or not does not play any significant role in the manuscript. The authors' key discovery here is the identification of an important mechanism of TMZ resistance, which is the acquisition of an enhancer element at the MGMT locus. This discovery is independent of whether there is a slight global decrease of chromatin marks at enhancers.

"We found the global levels of H3K4me3 and H3K9ac at promoters and H3K36me3 at gene bodies were similar between 5199 and 3080. Interestingly, although H3K4me1 and H3K27ac levels were quite similar between these two sub-lines based on Western blot analysis (Fig. 1b), their

enrichment at enhancer regions was reduced in the 3080 line compared to the 5199 line (Fig. 1d-e)."

I propose the authors just be upfront and replace this with e.g.:

"We found that global levels of H3K4me3, H3K9ac, H3K36me3, H3K4me1 and H3K27ac were largely similar between 5199 and 3080. We observed a slight reduction of H3K4me1 and H3K27ac at enhancers in the 3080 line compared to the 5199 line, though this difference needs to be corroborated by further tests using spiked-in internal controls (Fig. 1d-e)."

Reviewer #3

1B) The authors perform luciferase reporter assays to demonstrate that the enhancer element they identify has enhancer reporter activity. The assays reveal that fragments of the region they identify display enhancer reporter activity in these assays, but not all of them. The authors interpret this as the following:

"These results suggest that this region can enhance gene transcription and the R7 region may act as an MGMT enhancer, whereas other regions may serve as enhancer for other genes such as Ki67." (line 150-152)

The latter interpretation is likely incorrect. What the data suggests is that the R7 region (and R1 and R10 based on the data in Figure 2C) display enhancer activity in the reporter assay, and I recommend that authors state this. The authors used the SV40 promoter in the reporter system, which is by its nature artificial to begin with, so the data does not inform on which gene promoters the enhancer activates *in vivo*. To do that, the authors would have needed to clone the MGMT (or Ki67) promoter in the reporter vector. In my view, rephrasing this sentence as noted above is sufficient.

Response: We agree with the reviewer that our reporter assay data is not sufficient to definitively show R7 region can serve as an enhancer for Ki67. We modified the text on page 8 as "These results suggest that R1, R7 and R10 regions can enhance gene transcription from the SV40 promoter".

Reviewer Response

As I wrote before, the key result here is that the fragments have enhancer activity in the luciferase assay, and I suggested the authors simply state that, without speculating on the endogenous target gene of the enhancer.

"These results suggest that R1, R7 and R10 regions can enhance gene transcription from the SV40 promoter".

I propose the authors simplify this to be:

"These results suggest that R1, R7 and R10 regions have enhancer activity in luciferase reporter assays."

Reviewer #3

2) The data implicates the enhancer element discovered by the authors in the control of the MGMT and Ki67 genes, but whether the enhancer controls MGMT only or both MGMT and Ki67 is unclear. There is much emerging evidence that chromosome structure plays important roles in constraining enhancers to operate on specific genes. The genome is organized into CTCF-CTCF loops, that contribute to larger Topologically Associating Domains (TADs). Enhancers that occur within such CTCF-CTCF loops or TADs predominantly activate genes within the same loop/TAD. It would be very valuable to observe the 3D organization of the Ki67-enhancer-MGMT locus, which would inform on the likely physiological targets of the enhancer. Getting chromatin contact data in these cells would be quite complex, and in my view not necessary for this study, but the authors can explore published reference Hi-C (Rao et al, Cell 2014) or ChIA-PET datasets (Ji et al, Cell Stem Cell 2016, Tang et al, Cell 2015) for this purpose.

Response: We followed the reviewer's suggestion and examined several Hi-C datasets. We found that in neuroblastoma cells, Ki67, K-M enhancer and MGMT are located at the same TAD (Fig. 6b).

In cortical plate, Ki67 is located at the right boundary of TADs that contain MGMT and K-M enhancer (Fig. 6c). These results suggest that the enhancer can regulate the expression of MGMT in most cell lines, and may also regulate Ki67 in a cell type dependent manner. This finding is consistent with the differential effect of deletion of the K-M enhancer on the expression of MGMT and Ki67, suggesting that this enhancer primarily regulates MGMT expression. We included this result in Fig. 6 and make these points clear in the revised manuscript on page 14 as follow: "MKI67 is a gene encoding the nuclear protein Ki67, which is a proliferation marker for many tumors including GBM (Supplementary Fig. 8). Because MKI67 is another gene localized close to the enhancer, we first analyzed the published Hi-C datasets³⁸ to determine whether the MKI67 gene localized in the same topologically associating domain (TAD) with MGMT. We found that MKI67 resided in the same TAD as MGMT and the enhancer in neuroblastoma cells (Fig. 6b) and localized at the left boundary of the TAD domain containing MGMT in cortical and subcortical plate cells (Fig. 6c), suggesting that the K-M enhancer may also regulate Ki67 in a cell type dependent manner. We then analyzed Ki67 expression in the enhancer deletion clones. We observed that two SKMG3 clones with the 3.3 kb deletion, which show reduced proliferation (Fig. 6d), exhibited a significant decrease in Ki67 expression (Fig. 6e). Tumor growth rate and Ki67 expression were also reduced in the 3080 Del1 deleted clone (Fig. 6f-g). However, we noticed that in all these clones with the deletion of the enhancer element, the expression of MGMT was affected more than Ki67. Collectively, our results indicate that the K-M enhancer is the enhancer for MGMT that likely also regulates the expression of Ki67 in some cell type, which in turn regulates TMZ sensitivity and possibly cell proliferation."

Reviewer Response

The Hi-C data is a great addition to this paper, as it provides support that the endogenous target of the enhancer that authors described may indeed be both MKI67 and MGMT. When it comes to the interpretation of the Hi-C data, one needs to keep in mind the resolution of the data. Based on Figure 6b and 6c, I in fact get the impression based on the raw data that MKI67, the enhancer, and MGMT are in the same TAD both in the neuroblastoma and the cortical plate samples, but the annotation of the TAD boundary is not accurate because of the resolution of the Hi-C data (indeed there are gaps between where the TAD boundaries are annotated). So instead of speculating, the authors could simply state that they found evidence in Hi-C data that MKI67, the enhancer, and MGMT may be in the same TAD.

3) The authors propose that combination therapy with TMZ and transcription inhibitors may suppress the emergence of TMZ-resistance in glioblastoma. In addition to the BET inhibitors they mention, the authors may want to reference and suggest using CDK7 inhibitors to suppress emergence of drug resistance, as demonstrated in a recent study (Rusan et al, Cancer Discovery 2018).

Response: We have expanded the discussion to include this interesting study on page 16-17.

4) Typos:

Line 117: Moreover

Line 185: PDX

Line 325: Idea

Response: Those typos are corrected in the current manuscript.

Response to referees:

Reviewers' comments:

Reviewer #1(Remarks to the Author):

The authors have satisfactorily replied to my questions. The manuscript is improved and is now ready for publication.

Response: We thank the reviewer for the final comment.

Reviewer #2 (Remarks to the Author):

Although the authors have addressed the previous concerns, there are two areas that remain or have been created in the revision.

Response: We thank the reviewer for the very detailed comments about this manuscript. We have addressed each concern of the reviewer as detailed below.

1. line 52- the text suggests that the association between MGMT promoter methylation and favorable TMZ outcome was the basis for the development of MGMT inhibitors. This is not accurate as O6BG was developed long before the MGMT gene and promoter were even cloned. This inaccuracy should be corrected.

Response: To address this concern, we modified the sentence on line 54 as the following:

“However, combinations of TMZ with MGMT inhibitors such as O6-benzylguanine (O6BG), a synthetic derivative of guanine that can inhibit MGMT but was developed before the clone of the MGMT gene, resulted in enhanced hematologic toxicities, a reduced therapeutic window and no clinical benefit compared to TMZ alone^{10,11}.”

2. lines 300 and 373 - text added seems to imply that Ki67 expression is a driver of proliferation when existing data suggests that it is merely a marker of proliferation. If the authors wish to contend that Ki67 drives proliferation, additional studies are requires. Otherwise the text needs to be modified to eliminate the suggestion of causality.

Response: We appreciate the reviewer’s comments. We do not intend to link the expression of Ki67 with proliferation. In revised manuscript, we edit the text to reflect the fact that the proliferation of enhancer deleted tumor cells is reduced.

Line 314-315: “Collectively, our results indicate that the K-M enhancer is the enhancer for both MGMT and Ki67, which in turn regulates TMZ sensitivity and possibly cell proliferation.”

Line 378-379: we edited the original draft from “This enhancer likely also regulates the expression of Ki67 to regulate tumor proliferation” to “This enhancer likely also regulates the expression of Ki67as well as tumor proliferation.”

Reviewer #3 (Remarks to the Author):

The authors seemed to have missed the point on one key comment. I also have suggestions to improve the revisions the authors made based on the other comments. Below are the responses pasted in their rebuttal letter.

Response: We thank the reviewer for suggestions. We edited our manuscript accordingly as detailed below.

Reviewer #3 (Remarks to the Author):

Chen et al report the discovery of an enhancer element that controls the expression of MGMT that plays important roles in conferring resistance to the first line therapeutic agent TMZ in glioblastoma. The authors created maps of enhancers in TMZ-resistant and TMZ-sensitive glioblastoma cells using ChIP-Seq, and identified a region occupied by the enhancer-associated histone mark H3K27Ac specifically in TMZ-resistant cells upstream of the MGMT gene. H3K27Ac enrichment at this site correlates with MGMT expression and TMZ resistance in clinical specimens. Deletion of the enhancer in TMZ-resistant cells compromises MGMT expression and renders cells sensitive to TMZ. Activation of the enhancer in TMZ-sensitive cells induces MGMT expression and confers TMZ resistance. The study reports an important mechanism of acquired drug resistance in tumor cells, and suggests new therapeutic approaches to combat resistance through transcriptional inhibition. The experiments appear thoroughly designed and executed. I have a few suggestions to further improve the clarity and interpretation of some of the findings.

Response: We thank the reviewer’s very positive comments about our manuscript. We have addressed all of the reviewer’s minor concerns as detailed below.

Reviewer #3

Minor comments

1) The authors make a compelling case that the region they identified is indeed an enhancer element that controls MGMT expression. Their interpretation of some of the data at a few instances is slightly inaccurate, which is distracting to the readers. Below I list these instances and offer some

1A) The authors create ChIP-Seq maps of various histone modifications in TMZ-sensitive and TMZ-resistant cells, and then claim to observe global reduction in the signal intensity of some of these marks.

(E.g. “Interestingly the levels of H3K4me1 and H3K27ac at the enhancer regions were globally reduced in the 3080 line compared to the 5199 line. Moreover, the level of H3K9me3 was increased at the transcription starting sites (TSS) in the TMZ-resistant 3080 compared to the TMZ-sensitive 5199 (Fig. S1).”

This type of analysis is only possible when performing ChIP-Seq using internal controls such as precise amounts of spiked-in fragmented chromatin of other species (ChIP-RX, Orlando et al, Cell Reports 2015). Without the spiked-in internal control, it is not possible to rule out that any change in global levels is not caused by different quality of ChIP in the different samples. This type of global analysis is not necessary for the authors, and I do not suggest to repeat the ChIP experiments using internal controls. Instead, the authors are advised to report the numbers of peaks they identify in the data, and the correlation of the read densities in those regions in the TMZ-sensitive and TMZ-resistance cells. This shall give a reliable and comprehensive view of the differences in the enhancer landscape in these cells

Response: We agree with the reviewer that it is necessary to use spike-in chromatin to compare two samples when there are large changes in histone modifications, which we have done previously for other studies (Fang et al Science 2016). However, H3K4me1 and H3K27ac not only present on enhancer regions but also present in other regions such as TSS regions, 3' UTR regions and 5' URT regions. Although we observed that H3K4me1 and H3K27ac are reduced at enhancer regions, they are not globally reduced based on Western blot analysis (new Figure 1b). Therefore, we believe some comment on global changes can be made without use of spike-in chromatin. In revised manuscript, we carefully point out these ideas on line 105. We also performed the analysis suggested by the reviewer. The number of peaks we identified is listed in Table S1. The correlation of the read densities are shown on Letter Figure 3.

Reviewer Response

The authors seem to have missed the point here. Spike-in controls in ChIP are essential to determine global changes in occupancy levels, and are especially important when changes are small. The new data and the data in the references the authors now cite in fact argue that the so called global changes the authors claim to observe in this manuscript are erroneous. The authors cite their previous work (Fang et al, Science 2016) as the standard for detecting global changes in histone marks. In that paper, the authors used spike-in normalization for ChIP, and the Western blots they performed were consistent with the global changes (i.e. a reduction of signal was observed on the Western blot, AND a reduction of signal was observed in ChIP when accounting for the spike-in controls). In this paper, the authors do not use spike-ins, AND the Western blot (new Fig 1b) in fact contradicts their idea of global changes, (i.e. the levels of histone marks are unaffected).

At this point, I suggest that the authors forgo any speculation about global changes, as the experimental evidence for it is insufficient, and whether the changes are global or not does not play any significant role in the manuscript. The authors' key discovery here is the identification of an important mechanism of TMZ resistance, which is the acquisition of an

enhancer element at the MGMT locus. This discovery is independent of whether there is a slight global decrease of chromatin marks at enhancers.

“We found the global levels of H3K4me3 and H3K9ac at promoters and H3K36me3 at gene bodies were similar between 5199 and 3080. Interestingly, although H3K4me1 and H3K27ac levels were quite similar between these two sub-lines based on Western blot analysis (Fig. 1b), their enrichment at enhancer regions was reduced in the 3080 line compared to the 5199 line (Fig. 1d-e).”

I propose the authors just be upfront and replace this with e.g.:

“We found that global levels of H3K4me3, H3K9ac, H3K36me3, H3K4me1 and H3K27ac were largely similar between 5199 and 3080. We observed a slight reduction of H3K4me1 and H3K27ac at enhancers in the 3080 line compared to the 5199 line, though this difference needs to be corroborated by further tests using spiked-in internal controls (Fig. 1d-e).”

Response: We followed the reviewer’s suggestions and edited the sentence on line 108-111 as the following:

“Interestingly, although global H3K4me1 and H3K27ac levels were quite similar between these two sub-lines based on Western blot analysis (Fig. 1b), their enrichment at enhancer regions was reduced in the 3080 line compared to the 5199 line, though this difference needs to be corroborated by further tests using spiked-in internal controls (Fig. 1d-e).”

Reviewer #3

1B) The authors perform luciferase reporter assays to demonstrate that the enhancer element they identify has enhancer reporter activity. The assays reveal that fragments of the region they identify display enhancer reporter activity in these assays, but not all of them. The authors interpret this as the following:

“These results suggest that this region can enhance gene transcription and the R7 region may act as an MGMT enhancer, whereas other regions may serve as enhancer for other genes such as Ki67.” (line 150-152)

The latter interpretation is likely incorrect. What the data suggests is that the R7 region (and R1 and R10 based on the data in Figure 2C) display enhancer activity in the reporter assay, and I recommend that authors state this. The authors used the SV40 promoter in the reporter system, which is by its nature artificial to begin with, so the data does not inform on which gene promoters the enhancer activates in vivo. To do that, the authors would have needed to clone the MGMT (or Ki67) promoter in the reporter vector. In my view, rephrasing this sentence as noted above is sufficient.

Response: We agree with the reviewer that our reporter assay data is not sufficient to definitively show R7 region can serve as an enhancer for Ki67. We modified the text on page 8 as “These results suggest that R1, R7 and R10 regions can enhance gene transcription from the SV40 promoter”.

Reviewer Response

As I wrote before, the key result here is that the fragments have enhancer activity in the luciferase assay, and I suggested the authors simply state that, without speculating on the endogenous target gene of the enhancer.

“These results suggest that R1, R7 and R10 regions can enhance gene transcription from the SV40 promoter”.

I propose the authors simplify this to be:

“These results suggest that R1, R7 and R10 regions have enhancer activity in luciferase reporter assays.”

Response: We followed the reviewer’s suggestion and edited the sentence on line 160 as follow:

“These results suggest that R1, R7 and R10 regions have enhancer activity in luciferase reporter assays”

Reviewer #3

2) The data implicates the enhancer element discovered by the authors in the control of the MGMT and Ki67 genes, but whether the enhancer controls MGMT only or both MGMT and Ki67 is unclear. There is much emerging evidence that chromosome structure plays important roles in constraining enhancers to operate on specific genes. The genome is organized into CTCF-CTCF loops, that contribute to larger Topologically Associating Domains (TADs). Enhancers that occur within such CTCF-CTCF loops or TADs predominantly activate genes within the same loop/TAD. It would be very valuable to observe the 3D organization of the Ki67-enhancer-MGMT locus, which would inform on the likely physiological targets of the enhancer. Getting chromatin contact data in these cells would be quite complex, and in my view not necessary for this study, but the authors can explore published reference Hi-C (Rao et al, Cell 2014) or ChIA-PET datasets (Ji et al, Cell Stem Cell 2016, Tang et al, Cell 2015) for this purpose.

Response: We followed the reviewer’s suggestion and examined several Hi-C datasets. We found that in neuroblastoma cells, Ki67, K-M enhancer and MGMT are located at the same TAD (Fig. 6b). In cortical plate, Ki67 is located at the right boundary of TADs that contain MGMT and K-M enhancer (Fig. 6c). These results suggest that the enhancer can regulate the expression of MGMT in most cell lines, and may also regulate Ki67 in a cell type dependent manner. This finding is consistent with the differential effect of deletion of the K-M enhancer on the expression of MGMT and Ki67, suggesting that this enhancer primarily regulates MGMT expression. We included this result in Fig. 6 and make these points clear in the revised manuscript on page 14 as follow:

“MKI67 is a gene encoding the nuclear protein Ki67, which is a proliferation marker for many tumors including GBM (Supplementary Fig. 8). Because MKI67 is another gene localized close to the enhancer, we first analyzed the published Hi-C datasets³⁸ to determine whether the MKI67 gene localized in the same topologically associating domain (TAD) with

MGMT. We found that MKI67 resided in the same TAD as MGMT and the enhancer in neuroblastoma cells (Fig. 6b) and localized at the left boundary of the TAD domain containing MGMT in cortical and subcortical plate cells (Fig. 6c), suggesting that the K-M enhancer may also regulate Ki67 in a cell type dependent manner. We then analyzed Ki67 expression in the enhancer deletion clones. We observed that two SKMG3 clones with the 3.3 kb deletion, which show reduced proliferation (Fig. 6d), exhibited a significant decrease in Ki67 expression (Fig. 6e). Tumor growth rate and Ki67 expression were also reduced in the 3080 Dell deleted clone (Fig. 6f-g). However, we noticed that in all these clones with the deletion of the enhancer element, the expression of MGMT was affected more than Ki67. Collectively, our results indicate that the K-M enhancer is the enhancer for MGMT that likely also regulates the expression of Ki67 in some cell type, which in turn regulates TMZ sensitivity and possibly cell proliferation.”

Reviewer Response

The Hi-C data is a great addition to this paper, as it provides support that the endogenous target of the enhancer that authors described may indeed be both MKI67 and MGMT. When it comes to the interpretation of the Hi-C data, one needs to keep in mind the resolution of the data. Based on Figure 6b and 6c, I in fact get the impression based on the raw data that MKI67, the enhancer, and MGMT are in the same TAD both in the neuroblastoma and the cortical plate samples, but the annotation of the TAD boundary is not accurate because of the resolution of the Hi-C data (indeed there are gaps between where the TAD boundaries are annotated). So instead of speculating, the authors could simply state that they found evidence in Hi-C data that MKI67, the enhancer, and MGMT may be in the same TAD.

Response: We followed the reviewer’s suggestion and further edited our description on line 293-297 as follow:

“Since the TAD boundaries are not well defined, these results suggest that the K-M enhancer, MKI67 and MGMT may be in the same TAD. We then analyzed Ki67 expression in the enhancer deletion clones. We observed that two SKMG3 clones with the 3.3 kb deletion exhibited a significant decrease of Ki67 expression as well as reduced proliferation (Fig. 6d-e).”